# Extracting Reward Functions from Diffusion Models

**Felipe Nuti**[*]  **Tim Franzmeyer**[*]  **João F. Henriques**
`{nuti, frtim, joao}@robots.ox.ac.uk`

University of Oxford

## Abstract

Diffusion models have achieved remarkable results in image generation, and have similarly been used to learn high-performing policies in sequential decision-making tasks. Decision-making diffusion models can be trained on lower-quality data, and then be steered with a reward function to generate near-optimal trajectories. We consider the problem of extracting a reward function by comparing a decision-making diffusion model that models low-reward behavior and one that models high-reward behavior; a setting related to inverse reinforcement learning. We first define the notion of a *relative reward function of two diffusion models* and show conditions under which it exists and is unique. We then devise a practical learning algorithm for extracting it by aligning the gradients of a reward function – parametrized by a neural network – to the difference in outputs of both diffusion models. Our method finds correct reward functions in navigation environments, and we demonstrate that steering the base model with the learned reward functions results in significantly increased performance in standard locomotion benchmarks. Finally, we demonstrate that our approach generalizes beyond sequential decision-making by learning a reward-like function from two large-scale image generation diffusion models. The extracted reward function successfully assigns lower rewards to harmful images.[1]

## 1  Introduction

Recent work [25, 2] demonstrates that diffusion models – which display remarkable performance in image generation – are similarly applicable to sequential decision-making. Leveraging a well-established framing of reinforcement learning as conditional sampling [37], Janner et al. [25] show that diffusion models can be used to parameterize a reward-agnostic prior distribution over trajectories, learned from offline demonstrations alone. Using classifier guidance [61], the diffusion model can then be steered with a (cumulative) reward function, parametrized by a neural network, to generate near-optimal behaviors in various sequential decision-making tasks. Hence, it is possible to learn successful policies both through (a) training an *expert diffusion model* on a distribution of optimal trajectories, and (b) by training a *base diffusion model* on lower-quality or reward-agnostic trajectories and then steering it with the given reward function. This suggests that the reward function can be extracted by comparing the distributions produced by such base and expert diffusion models.

The problem of learning the preferences of an agent from observed behavior, often expressed as a reward function, is considered in the Inverse Reinforcement Learning (IRL) literature [56, 43]. In contrast to merely imitating the observed behavior, learning the reward function behind it allows for better robustness, better generalization to distinct environments, and for combining rewards extracted from multiple behaviors.

---

[*]Equal Contribution.

[1]Video and Code at `https://www.robots.ox.ac.uk/~vgg/research/reward-diffusion/`

37th Conference on Neural Information Processing Systems (NeurIPS 2023).

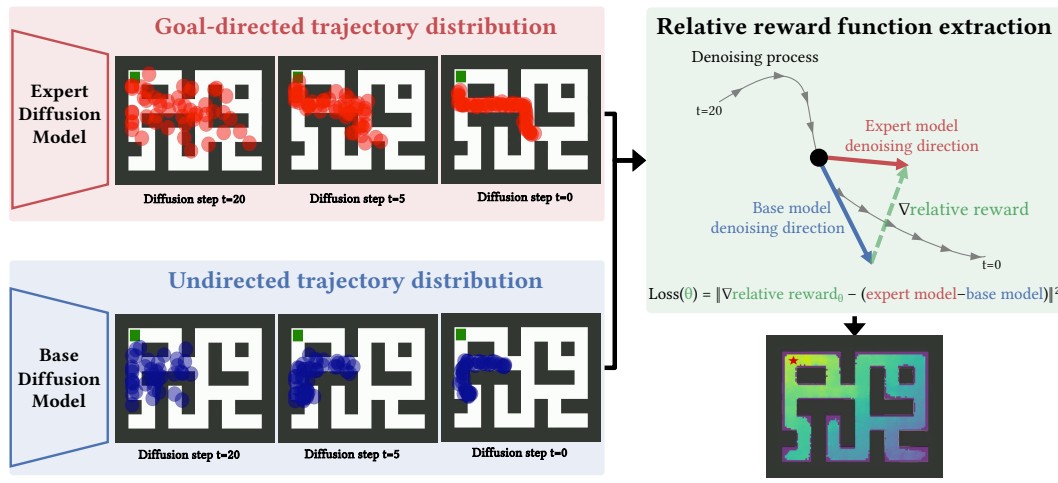

Figure 1: We see 2D environments with black walls, in which an agent has to move through the maze to reach the goal in the top left corner (green box). The red shaded box shows the progression from an initial noised distribution over states (at diffusion timestep $t = 20$, left) to a denoised high-reward expert trajectory on the right. This distribution is modeled by an *expert diffusion model*. The blue shaded box depicts the same process but for a low-reward trajectory where the agent moves in the wrong direction. This distribution is modeled by a *base diffusion model*. Our method (green shaded box) trains a neural network to have its gradient aligned to the difference in outputs of these two diffusion models throughout the denoising process. As we argue in Section 4, this allows us to extract a relative reward function of the two models. We observe that the heatmap of the learned relative reward (bottom right) assigns high rewards to trajectories that reach the goal point (red star).

Prior work in reward learning is largely based on the Maximum Entropy Inverse Reinforcement Learning (MaxEntIRL) framework [71], which relies on alternating between policy optimization and reward learning. This often comes with the assumption of access to the environment (or a simulator) to train the policy. Reward learning is also of independent interest outside of policy learning, for understanding agents' behavior or predicting their actions, with applications in Value Alignment, interpretability, and AI Safety [3, 14, 55]. For example, reward functions can also be utilized to better understand an existing AI system's explicit or implicit "preferences" and tendencies [57].

In this work, we introduce a method for extracting a relative reward function from two decision-making diffusion models, illustrated in Figure 1. Our method does not require environment access, simulators, or iterative policy optimization. Further, it is agnostic to the architecture of the diffusion models used, applying to continuous and discrete models, and making no assumption of whether the models are unguided, or whether they use either classifier guidance [61] or classifier-free guidance [22].

We first derive a notion of a *relative reward function of two diffusion models*. We show that, under mild assumptions on the trajectory distribution and diffusion sampling process, our notion of reward exists and is unique, up to an additive constant. Further, we show that the derived reward is equivalent to the true reward under the probabilistic RL framework [37]. Finally, we propose a practical learning algorithm for extracting the relative reward function of two diffusion models by aligning the gradients of the learned reward function with the *differences of the outputs* of the base and expert models. As the extracted reward function itself is a feed-forward neural network, i.e. not a diffusion model, it is computationally lightweight.

Our proposed method for extracting a relative reward function of two diffusion models could be applicable to several scenarios. For example, it allows for better interpretation of behavior differences, for composition and manipulation of reward functions, and for training agents from scratch or fine-tune existing policies. Further, relative reward functions could allow to better understand diffusion models by contrasting them. For example, the biases of large models trained on different datasets are not always obvious, and our method may aid interpretability and auditing of models by revealing the differences between the outputs they are producing.

We empirically evaluate our reward learning method along three axes. In the Maze2D environments [12], we learn a reward function by comparing a base diffusion model trained on exploratory trajectories and an expert diffusion model trained on goal-directed trajectories, as illustrated in Figure 1. We can see in Figure 3 that our method learns the correct reward function for varying maze configurations. In the common locomotion environments Hopper, HalfCheetah, and Walker2D [12, 7], we learn a reward function by comparing a low-performance base model to an expert diffusion model and demonstrate that steering the base model with the learned reward function results in a significantly improved performance. Beyond sequential-decision making, we learn a reward-like function by comparing a base image generation diffusion model (Stable Diffusion, [54]) to a *safer* version of Stable Diffusion [59]. Figure 2 shows that the learned reward function penalizes images with harmful content, such as violence and hate, while rewarding harmless images.

In summary, our work makes the following contributions:

• We introduce the concept of *relative reward functions* of diffusion models, and provide a mathematical analysis of their relation to rewards in sequential decision-making.
• We propose a practical learning algorithm for extracting relative reward functions by aligning the reward function's gradient with the difference in outputs of two diffusion models.
• We empirically validate our method in long-horizon planning environments, in high-dimensional control environments, and show generalization beyond sequential decision-making.

## 2   Related Work

*Diffusion models*, originally proposed by Sohl-Dickstein et al. [61], are an expressive class of generative models that generate samples by learning to invert the process of noising the data. The work of Ho et al. [21] led to a resurgence of interest in the method, followed by [44, 10, 62]. Song et al. [65] proposed a unified treatment of diffusion models and score-based generative models [63, 64] through stochastic differential equations [45], used e.g. in [32]. Diffusion models have shown excellent performance in image generation [54, 10, 59, 52], molecule generation [23, 26, 27], 3D generation [48, 41], video generation [20], language [38], and, crucially for this work, sequential decision-making [25]. Part of their appeal as generative models is due to their *steerability*. Since sample generation is gradual, other pre-trained neural networks can be used to steer the diffusion model during sampling (i.e. classifier guidance, Section 3). An alternative approach is classifier-free guidance [22], often used in text-to-image models such as [54].

Decision-making with diffusion models have first been proposed by Janner et al. [25], who presented a hybrid solution for planning and dynamics modeling by iteratively denoising a sequence of states and actions at every time step, and executing the first denoised action. This method builds on the probabilistic inference framework for RL, reviewed by Levine [37], based on works including [66, 70, 71, 31, 30]. Diffusion models have been applied in the standard reinforcement learning setting [39, 2, 8], and also in 3D domains [24]. They have similarly been used in imitation learning for generating more human-like policies [47, 53], for traffic simulation [69], and for offline reinforcement learning [25, 68, 17]. Within sequential-decision making, much interest also lies in extracting reward functions from observed behavior. This problem of inverse RL [1, 5] has mostly either been approached by making strong assumptions about the structure of the reward function [71, 43, 15], or, more recently, by employing adversarial training objectives [13].

The optimization objective of our method resembles that of physics-informed neural networks [6, 16, 40, 50, 51, 11], which also align neural network gradients to a pre-specified function. In our case, this function is a difference in the outputs of two diffusion models.

To the best of our knowledge, no previous reward learning methods are directly applicable to diffusion models, nor has extracting a relative reward function from two diffusion models been explored before.

## 3   Background

Our method leverages mathematical properties of diffusion-based planners to extract reward functions from them. To put our contributions into context and define notation, we give a brief overview of the probabilistic formulation of Reinforcement Learning presented in [37], then of diffusion models, and finally of how they come together in decision-making diffusion models [25].

**Reinforcement Learning as Probabilistic Inference.** Levine [37] provides an in-depth exposition of existing methods for approaching sequential decision-making from a probabilistic and causal angle using Probabilistic Graphical Models (PGM) [28]. We review some essential notions to understand this perspective on RL. Denote by $\Delta_{\mathcal{X}}$ the set of probability distributions over a set $\mathcal{X}$.

**Markov Decision Process (MDP).** An MDP is a tuple $\langle \mathcal{S}, \mathcal{A}, \mathcal{T}, r, \rho_0, T \rangle$ consisting of a state space $\mathcal{S}$, an action space $\mathcal{A}$, a transition function $\mathcal{T} : \mathcal{S} \times \mathcal{A} \to \Delta_{\mathcal{S}}$, a reward function $r : \mathcal{S} \times \mathcal{A} \to \Delta_{\mathbb{R}_{\geq 0}}$, an initial state distribution $\rho_0 \in \Delta_{\mathcal{S}}$, and an episode length $T$. The MDP starts at an initial state $s_0 \sim \rho_0$, and evolves by sampling $a_t \sim \pi(s_t)$, and then $s_{t+1} \sim \mathcal{T}(s_t, a_t)$ for $t \geq 0$. The reward received at time $t$ is $r_t \sim r(s_t, a_t)$. The episode ends at $t = T$. The sequence of state-action pairs $\tau = ((s_t, a_t))_{t=0}^T$ is called a *trajectory*.

The framework in [37] recasts such an MDP as a PGM, illustrated in Appendix B, Figure 5b. It consists of a sequence of states $(s_t)_{t=0}^T$, actions $(a_t)_{t=0}^T$ and *optimality variables* $(\mathcal{O}_t)_{t=0}^T$. The reward function $r$ is not explicitly present. Instead, it is encoded in the optimality variables via the relation: $\mathcal{O}_t \sim \text{Ber}(e^{-r(s_t, a_t)})$. One can apply Bayes's Rule to obtain:

$$p(\tau | \mathcal{O}_{1:T}) \propto p(\tau) \cdot p(\mathcal{O}_{1:T} | \tau) \tag{1}$$

which factorizes the distribution of optimal trajectories (up to a normalizing constant) as a *prior* $p(\tau)$ over trajectories and a *likelihood term* $p(\mathcal{O}_{1:T} | \tau)$. From the definition of $\mathcal{O}_{1:T}$ and the PGM structure, we have $p(\mathcal{O}_{1:T} | \tau) = e^{-\sum_t r(s_t, a_t)}$. Hence, the negative log-likelihood of optimality conditioned on a trajectory $\tau$ corresponds to its cumulative reward: $-\log p(\mathcal{O}_{1:T} | \tau) = \sum_t r(s_t, a_t)$.

**Diffusion Models in Continuous Time.** At a high level, diffusion models work by adding noise to data $\mathbf{x} \in \mathbb{R}^n$ (*forward process*), and then learning to denoise it (*backward process*).

The forward noising process in continuous time follows the Stochastic Differential Equation (SDE):

$$d\mathbf{x}_t = \mathbf{f}(\mathbf{x}_t, t)dt + g(t)d\mathbf{w}_t \tag{2}$$

where $f$ is a function that is Lipschitz, $\mathbf{w}$ is a standard Brownian Motion [36] and $g$ is a (continuous) noise schedule, which regulates the amount of noise added to the data during the forward process (c.f. A.2). Song et al. [65] then use a result of Anderson [4] to write the SDE satisfied by the *reverse process* of (2), denoted $\bar{\mathbf{x}}_t$, as:

$$d\bar{\mathbf{x}}_t = \left[\mathbf{f}(\bar{\mathbf{x}}_t, t) - g(t)^2 \nabla_{\mathbf{x}} \log p_t(\bar{\mathbf{x}}_t)\right] dt + g(t) \, d\bar{\mathbf{w}}_t \tag{3}$$

Here $p_t(\mathbf{x})$ denotes the marginal density function of the forward process $\mathbf{x}_t$, and $\bar{\mathbf{w}}$ is a reverse Brownian motion (see [4]). The diffusion model is a neural network $\mathbf{s}_\Theta(\mathbf{x}_t, t)$ with parameters $\Theta$ that is trained to approximate $\nabla_{\mathbf{x}} \log p_t(\mathbf{x}_t)$, called the *score* of the distribution of $\mathbf{x}_t$. The network $\mathbf{s}_\Theta(\mathbf{x}_t, t)$ can then be used to generate new samples from $p_0$ by taking $\bar{\mathbf{x}}_T \sim \mathcal{N}(0, I)$ for some $T > 0$, and simulating (3) *backwards in time* to arrive at $\bar{\mathbf{x}}_0 \approx \mathbf{x}_0$, with the the neural network $\mathbf{s}_\Theta$ in place of the score term:

$$d\bar{\mathbf{x}}_t = \left[\mathbf{f}(\bar{\mathbf{x}}_t, t) - g(t)^2 \mathbf{s}_\Theta(\bar{\mathbf{x}}_t, t)\right] dt + g(t) \, d\bar{\mathbf{w}}_t \tag{4}$$

This formulation is essential for deriving existence results for the relative reward function of two diffusion models in Section 4, as it allows for conditional sampling. For example, to sample from $p(\mathbf{x}_0 | y) \propto p(\mathbf{x}_0) \cdot p(y | \mathbf{x}_0)$, where $y \in \{0, 1\}$, the sampling procedure can be modified to use $\nabla_{\mathbf{x}} \log p(\mathbf{x}_t | y) \approx \mathbf{s}_\Theta(\mathbf{x}_t, t) + \nabla_{\mathbf{x}} \rho(\mathbf{x}, t)$ instead of $\mathbf{s}_\Theta(\bar{\mathbf{x}}_t, t)$. Here, $\rho(\mathbf{x}, t)$ is a neural network approximating $\log p(y | \mathbf{x}_t)$. The gradients of $\rho$ are multiplied by a small constant $\omega$, called the *guidance scale*. The resulting *guided* reverse SDE is as follows:

$$d\bar{\mathbf{x}}_t = \left[\mathbf{f}(\bar{\mathbf{x}}_t, t) - g(t)^2[\mathbf{s}_\Theta(\bar{\mathbf{x}}_t, t) + \omega \nabla_{\mathbf{x}} \rho(\bar{\mathbf{x}}_t, t)]\right] dt + g(t) \, d\bar{\mathbf{w}}_t \tag{5}$$

Informally, this method, often called *classifier guidance* [61], allows for steering a diffusion model to produce samples $\mathbf{x}$ with some property $y$ by gradually pushing the samples in the direction that maximizes the output of a classifier predicting $p(y | \mathbf{x})$.

**Planning with Diffusion.** The above shows how sequential decision-making can be framed as sampling from a posterior distribution $p(\tau | \mathcal{O}_{1:T})$ over trajectories. Section 3 shows how a diffusion model $p(\mathbf{x}_0)$ can be combined with a classifier to sample from a posterior $p(\mathbf{x}_0 | y)$. These two observations point us to the approach in Diffuser [25]: using a diffusion model to model a prior $p(\tau)$ over trajectories, and a reward prediction function $\rho(\mathbf{x}, t) \approx p(\mathcal{O}_{1:T} | \tau_t)$ to steer the diffusion model. This allows approximate sampling from $p(\tau | \mathcal{O}_{1:T})$, which produces (near-)optimal trajectories.

The policy generated by the Diffuser denoises a fixed-length sequence of future states and actions and executes the first action of the sequence. Diffuser can also be employed for goal-directed planning, by fixing initial and goal states during the denoising process.

## 4 Methods

In the previous section, we established the connection between the (cumulative) return $\sum_t r(s_t, a_t)$ of a trajectory $\tau = ((s_t, a_t))_{t=1}^T$ and the value $p(y|\mathbf{x})$ used for classifier guidance (Section 3), with $\mathbf{x}$ corresponding to $\tau$ and $y$ corresponding to $\mathcal{O}_{1:T}$. Hence, we can look at our goal as finding $p(y|\mathbf{x})$, or equivalently $p(\mathcal{O}_{1:T}|\tau)$, in order to recover the cumulative reward for a given trajectory $\tau$. Additionally, in Section 4.3, we will demonstrate how the single-step reward can be computed by choosing a specific parametrization for $p(y|\mathbf{x})$.

**Problem Setting.** We consider a scenario where we have two decision-making diffusion models: a base model $\mathbf{s}_\phi^{(1)}$ that generates reward-agnostic trajectories, and an expert model $\mathbf{s}_\Theta^{(2)}$ that generates trajectories optimal under some *unknown* reward function $r$. Our objective is to learn a reward function $\rho(\mathbf{x}, t)$ such that, if $\rho$ is used to steer the base model $\mathbf{s}_\phi^{(1)}$ through classifier guidance, we obtain a distribution close to that of the expert model $\mathbf{s}_\Theta^{(2)}$. From the above discussion, such a function $\rho$ would correspond to the notion of relative reward in the probabilistic RL setting.

In the following, we present theory showing that:

**1.** In an idealized setting where $\mathbf{s}^{(1)}$ and $\mathbf{s}^{(2)}$ have no approximation error (and are thus conservative vector fields), there exists a unique function $\rho$ that exactly converts $\mathbf{s}^{(1)}$ to $\mathbf{s}^{(2)}$ through classifier guidance.
**2.** In practice, we cannot expect such a classifier to exist, as approximation errors might result in diffusion models corresponding to non-conservative vector fields.
**3.** However, the functions $\rho$ that *best approximate* the desired property (to arbitrary precision $\varepsilon$) do exist. These are given by Def. 4.4 and can be obtained through a *projection* using an $L^2$ distance.
**4.** The use of an $L^2$ distance naturally results in an $L^2$ loss for learning $\rho$ with Gradient Descent.

### 4.1 A Result on Existence and Uniqueness

We now provide a result saying that, once $\mathbf{s}_\phi^{(1)}$ and $\mathbf{s}_\Theta^{(2)}$ are fixed, there is a condition on the gradients of $\rho$ that, if met, would allow us to match not only the *distributions* of $\mathbf{s}_\phi^{(1)}$ and $\mathbf{s}_\Theta^{(2)}$, but also their entire denoising processes, with probability 1 (i.e. *almost surely*, a.s.).

In the following theorem, $\mathbf{h}$ plays the role of the gradients $\nabla_\mathbf{x}\rho(\mathbf{x}_t, t)$. Going from time $t = 0$ to $t = T$ in the theorem corresponds to solving the backward SDE (3) from $t = T$ to $t = 0$, and $\mathbf{f}$ in the theorem corresponds to the drift term (i.e. coefficient of $dt$) of (3). For proof, see Appendix C.

**Theorem 4.1** (Existence and Uniqueness). *Let $T > 0$. Let $\mathbf{f}^{(1)}$ and $\mathbf{f}^{(2)}$ be functions from $\mathbb{R}^n \times [0, T]$ to $\mathbb{R}^n$ that are Lipschitz, and $g : [0, T] \to \mathbb{R}_{\geq 0}$ be bounded and continuous with $g(0) > 0$. Fix a probability space $(\Omega, \mathcal{F}, \mathbb{P})$, and a standard $\mathbb{R}^n$-Brownian Motion $(\mathbf{w}_t)_{t\geq 0}$.*

*Consider the Itô SDEs:*

$$d\mathbf{x}_t^{(1)} = \mathbf{f}^{(1)}(\mathbf{x}_t^{(1)}, t)\, dt + g(t)\, d\mathbf{w}_t \tag{6}$$

$$d\mathbf{x}_t^{(2)} = \mathbf{f}^{(2)}(\mathbf{x}_t^{(2)}, t)\, dt + g(t)\, d\mathbf{w}_t \tag{7}$$

$$d\mathbf{x}_t\ \ = [\mathbf{f}^{(1)}(\mathbf{x}_t, t) + \mathbf{h}(\mathbf{x}_t, t)]\, dt + g(t)\, d\mathbf{w}_t, \text{ where } \mathbf{h} \text{ is Lipschitz} \tag{8}$$

*and fix an initial condition $\mathbf{x}_0^{(1)} = \mathbf{x}_0^{(2)} = \mathbf{x}_0 = \mathbf{z}$, where $\mathbf{z}$ is a random variable with $\mathbb{E}[||\mathbf{z}||_2^2] < \infty$.*

*Then (6), (7), and (8) have almost surely (a.s.) unique solutions $\mathbf{x}^{(1)}$, $\mathbf{x}^{(2)}$ and $\mathbf{x}$ with a.s. continuous sample paths. Furthermore, there exists an a.s. unique choice of $\mathbf{h}$ such that $\mathbf{x}_t = \mathbf{x}_t^{(2)}$ for all $t \geq 0$, a.s., which is given by*

$$\mathbf{h}(\mathbf{x}, t) = \mathbf{f}^{(2)}(\mathbf{x}, t) - \mathbf{f}^{(1)}(\mathbf{x}, t). \tag{9}$$

In all diffusion model methods we are aware of, the drift and noise terms of the backward process indeed satisfy the pre-conditions of the theorem, under reasonable assumptions on the data distribution (see C.3), and using a network with smooth activation functions like Mish [42] or GeLU [19].

Therefore, Theorem 4.1 tells us that, if we were free to pick the gradients $\nabla_{\mathbf{x}}\rho(\mathbf{x}_t, t)$, setting them to $\mathbf{f}^{(2)}(\mathbf{x}_t, t) - \mathbf{f}^{(1)}(\mathbf{x}_t, t)$ would be the "best" choice: it is the only choice resulting in guided samples *exactly reproducing* the whole process $\mathbf{x}^{(2)}$ (and, in particular, the distribution of $\mathbf{x}_0^{(2)}$). We will now see that, in an idealized setting, there exists a unique function $\rho$ satisfying this criterion. We start by recalling the concept of a *conservative vector field*, from multivariate calculus, and how it relates to gradients of continuously differentiable functions.

**Definition 4.2** (Conservative Vector Field, Definition 7.6 in [35])**.** We say that a vector field $\mathbf{f}$ is conservative if it is the gradient of a continuously differentiable function $\Phi$,

$$\mathbf{f}(\mathbf{x}) = \nabla_{\mathbf{x}}\Phi(\mathbf{x})$$

for all $\mathbf{x}$ in the domain of $\mathbf{f}$. The function $\Phi$ is called a potential function of $\mathbf{f}$.

Suppose we had access to the ground-truth scores $\mathbf{s}_{\text{true}}^{(1)}(\mathbf{x}, t)$ and $\mathbf{s}_{\text{true}}^{(2)}(\mathbf{x}, t)$ of the forward processes for the base and expert models (i.e. no approximation error). Then they are equal to $\nabla_{\mathbf{x}}\log p_t^{(1)}(\mathbf{x})$ and $\nabla_{\mathbf{x}}\log p_t^{(2)}(\mathbf{x})$, respectively. If we also assume $p_t^{(1)}(\mathbf{x})$ and $p_t^{(2)}(\mathbf{x})$ are continuously differentiable, we have that, by Definition 4.2, the diffusion models are conservative for each $t$. Thus, their difference is also conservative, i.e. the gradient of a continuously differentiable function.

Hence, by the Fundamental Theorem for Line Integrals (Th. 7.2 in [35]), there exists a unique $\rho$ satisfying $\nabla_{\mathbf{x}}\rho(\mathbf{x}, t) = \mathbf{s}_{\text{true}}^{(2)}(\mathbf{x}, t) - \mathbf{s}_{\text{true}}^{(1)}(\mathbf{x}, t)$, up to an additive constant, given by the line integral

$$\rho(\mathbf{x}, t) = \int_{\mathbf{x}_{\text{ref}}}^{\mathbf{x}} [\mathbf{s}_{\text{true}}^{(2)}(\mathbf{x}', t) - \mathbf{s}_{\text{true}}^{(1)}(\mathbf{x}', t)] \cdot d\mathbf{x}', \tag{10}$$

where $\mathbf{x}_{\text{ref}}$ is some arbitrary reference point, and the line integral is path-independent.

In practice, however, we cannot guarantee the absence of approximation errors, nor that the diffusion models are conservative.

## 4.2 Relative Reward Function of Two Diffusion Models

To get around the possibility that $\mathbf{s}^{(1)}$ and $\mathbf{s}^{(2)}$ are not conservative, we may instead look for the conservative field best approximating $\mathbf{s}^{(2)}(\mathbf{x}, t) - \mathbf{s}^{(1)}(\mathbf{x}, t)$ in $L^2(\mathbb{R}^n, \mathbb{R}^n)$ (i.e. the space of square-integrable vector fields, endowed with the $L^2$ norm). Using a well-known fundamental result on uniqueness of projections in $L^2$ (Th. C.8), we obtain the following:

**Proposition 4.3** (Optimal Relative Reward Gradient)**.** *Let $\mathbf{s}_\phi^{(1)}$ and $\mathbf{s}_\Theta^{(2)}$ be any two diffusion models and $t \in (0, T]$, with the assumption that $\mathbf{s}_\Theta^{(2)}(\cdot, t) - \mathbf{s}_\phi^{(1)}(\cdot, t)$ is square-integrable. Then there exists a unique vector field $\mathbf{h}_t$ given by*

$$\mathbf{h}_t = \underset{\mathbf{f} \in \overline{\text{Cons}(\mathbb{R}^n)}}{\text{argmin}} \int_{\mathbb{R}^n} ||\mathbf{f}(\mathbf{x}) - (\mathbf{s}_\Theta^{(2)}(\mathbf{x}, t) - \mathbf{s}_\phi^{(1)}(\mathbf{x}, t))||_2^2 \, d\mathbf{x} \tag{11}$$

*where $\overline{\text{Cons}(\mathbb{R}^n)}$ denotes the closed span of gradients of smooth $W^{1,2}$ potentials. Furthermore, for any $\varepsilon > 0$, there is a smooth, square-integrable potential $\Phi$ with a square-integrable gradient satisfying:*

$$\int_{\mathbb{R}^n} ||\nabla_{\mathbf{x}}\Phi(\mathbf{x}) - \mathbf{h}_t(\mathbf{x})||_2^2 \, d\mathbf{x} < \varepsilon \tag{12}$$

*We call such an $\mathbf{h}_t$ the **optimal relative reward gradient of $\mathbf{s}_\phi^{(1)}$ and $\mathbf{s}_\Theta^{(2)}$ at time** $t$.*

For proof of Proposition 4.3 see Appendix C.4. It is important to note that for the projection to be well-defined, we required an assumption regarding the integrability of the diffusion models. Without this assumption, the integral would simply diverge.

The result in Proposition 4.3 tells us that we can get arbitrarily close to the optimal relative reward gradient using scalar potentials' gradients. Therefore, we may finally define the central notion in this paper:

**Algorithm 1:** Relative reward function training.

---

**Input:** Base $\mathbf{s}^{(1)}$ and expert $\mathbf{s}^{(2)}$ diffusion models, dataset $\mathcal{D}$,
      number of iterations $I$.
**Output:** Relative reward estimator $\rho_\theta$.
Initialize reward estimator parameters $\theta$.
**for** $j \in \{1, ..., I\}$ **do**
    | Sample batch: $\mathbf{X}_0 = [\mathbf{x}_0^{(1)}, ..., \mathbf{x}_0^{(N)}]$ from $\mathcal{D}$
    | Sample times: $\mathbf{t} = [t_1, ..., t_N]$ independently in $\mathcal{U}(0, T)$
    | Sample forward process: $\mathbf{X_t} \leftarrow [\mathbf{x}_{t_1}^{(1)}, ..., \mathbf{x}_{t_N}^{(N)}]$
    | Take an optimization step on $\theta$ according to $\hat{L}_{\mathrm{RRF}}(\theta) =$
    |    $\frac{1}{N} \sum_{i=1}^N ||\nabla_{\mathbf{x}} \rho_\theta(\mathbf{x}_{t_i}^{(i)}, t_i) - (\mathbf{s}_\Theta^{(2)}(\mathbf{x}_{t_i}^{(i)}, t_i) - \mathbf{s}_\phi^{(1)}(\mathbf{x}_{t_i}^{(i)}, t_i))||_2^2$
**end**

---

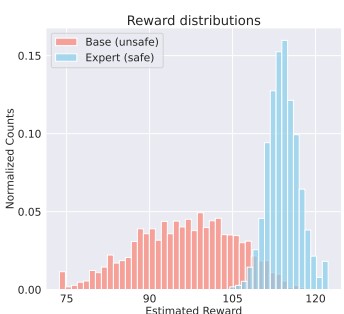

Figure 2: Learned rewards for base and expert diffusion models from Stable Diffusion (Sec. 5). Prompts from the I2P dataset.

**Definition 4.4** ($\varepsilon$-Relative Reward Function). For an $\varepsilon > 0$, an $\varepsilon$-relative reward function of diffusion models $\mathbf{s}_\phi^{(1)}$ and $\mathbf{s}_\Theta^{(2)}$ is a function $\rho : \mathbb{R}^n \times [0, T] \to \mathbb{R}$ such that

$$\forall t \in (0, T] : \int_{\mathbb{R}^n} ||\nabla_{\mathbf{x}} \rho(\mathbf{x}, t) - \mathbf{h}_t(\mathbf{x})||_2^2 \, d\mathbf{x} < \varepsilon \tag{13}$$

where $\mathbf{h}_t$ denotes the optimal relative reward gradient of $\mathbf{s}_\phi^{(1)}$ and $\mathbf{s}_\Theta^{(2)}$ at time $t$.

### 4.3 Extracting Reward Functions

We now set out to actually approximate the relative reward function $\rho$. Definition 4.4 naturally translates into an $L^2$ training objective for learning $\rho$:

$$L_{\mathrm{RRF}}(\theta) = \mathbb{E}_{t \sim \mathcal{U}[0,T], \mathbf{x}_t \sim p_t} \left[ ||\nabla_{\mathbf{x}} \rho_\theta(\mathbf{x}_t, t) - (\mathbf{s}_\Theta^{(2)}(\mathbf{x}_t, t) - \mathbf{s}_\phi^{(1)}(\mathbf{x}_t, t))||_2^2 \right] \tag{14}$$

where $p_t$ denotes the marginal at time $t$ of the forward noising process.

We optimize this objective via Empirical Risk Minimization and Stochastic Gradient Descent. See Algorithm 1 for a version assuming access to the diffusion models and their training datasets, and Algorithm 2 in Appendix D for one which does not assume access to any pre-existing dataset. Our method requires no access to the environment or to a simulator. Our algorithm requires computing a second-order mixed derivative $D_\theta(\nabla_{\mathbf{x}} \rho(\mathbf{x}, t)) \in \mathbb{R}^{m \times n}$ (where $m$ is the number of parameters $\theta$), for which we use automatic differentiation in PyTorch [46].

**Recovering Per-Time-Step Rewards.** We can parameterize $\rho$ using a single-time-step neural network $g_\theta(s, a, t)$ as $\rho(\tau_t, t) = \frac{1}{N} \sum_{i=1}^N g_\theta(s_t^i, a_t^i, t)$, where $N$ is the horizon and $\tau_t$ denotes a trajectory at diffusion timestep $t$, and $s_t^i$ and $a_t^i$ denote the $i$th state and action in $\tau_t$. Then, $g_\theta(s, a, 0)$ predicts a reward for a state-action pair $(s, a)$.

## 5 Experiments

In this section, we conduct empirical investigations to analyze the properties of relative reward functions in practice. Our experiments focus on three main aspects: the alignment of learned reward functions with the goals of the expert agent, the performance improvement achieved through steering the base model using the learned reward, and the generalizability of learning reward-like functions to domains beyond decision-making (note that the relative reward function is defined for any pair of diffusion models). For details on implementation specifics, computational requirements, and experiment replication instructions, we refer readers to the Appendix.

### 5.1 Learning Correct Reward Functions from Long-Horizon Plans

Maze2D [12] features various environments which involve controlling the acceleration of a ball to navigate it towards various goal positions in 2D mazes. It is suitable for evaluating reward learning, as it requires effective credit assignment over extended trajectories.

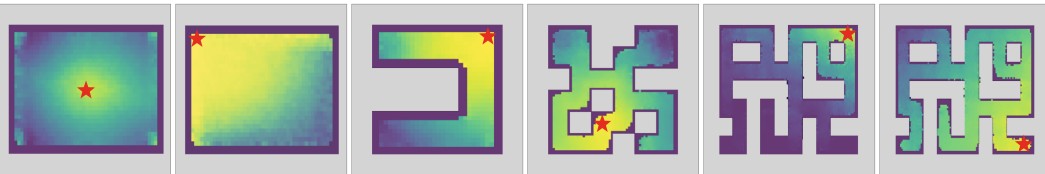

Figure 3: Heatmaps of the learned relative reward in Maze2D. ★ denotes the ground-truth goal position. For more examples, see Appendix E.3.

**Implementation.** To conduct our experiments, we generate multiple datasets of trajectories in each of the 2D environments, following the data generation procedure from D4RL [12], except sampling start and goal positions uniformly at random (as opposed to only at integer coordinates).

For four maze environments with different wall configurations (depicted in Figure 3), we first train a base diffusion model on a dataset of uniformly sampled start and goal positions, hence representing undirected behavior. For each environment, we then train eight expert diffusion models on datasets with fixed goal positions. As there are four maze configurations and eight goal positions per maze configuration, we are left with 32 expert models in total. For each of the 32 expert models, we train a relative reward estimator $\rho_\theta$, implemented as an MLP, via gradient alignment, as described in Alg. 1. We repeat this across 5 random seeds.

**Discriminability Results.** For a quantitative evaluation, we train a logistic regression classifier per expert model on a balanced dataset of base and expert trajectories. Its objective is to label trajectories as base or expert using only the predicted reward as input. We repeat this process for each goal position with 5 different seeds, and average accuracies across seeds. The achieved accuracies range from 65.33% to 97.26%, with a median of 84.49% and a mean of 83.76%. These results demonstrate that the learned reward effectively discriminates expert trajectories.

**Visualization of Learned Reward Functions.** To visualize the rewards, we use our model to predict rewards for fixed-length sub-trajectories in the base dataset. We then generate a 2D heatmap by averaging the rewards of all sub-trajectories that pass through each grid cell. Fig. 3 displays some of these heatmaps, with more examples given in App. F.

**Results.** We observe in Fig. 3 that the network accurately captures the rewards, with peaks occurring at the true goal position of the expert dataset in 78.75% ± 8.96% of the cases for simpler mazes (`maze2d-open-v0` and `maze2d-umaze-v0`), and 77.50% ± 9.15% for more advanced mazes (`maze2d-medium-v1` and `maze2d-large-v1`). Overall, the network achieves an average success rate of 78.12% ± 6.40%. We further verify the correctness of the learned reward functions by retraining agents in Maze2D using the extracted reward functions (see Appendix E.2.1).

**Sensitivity to Dataset Size.** Since training diffusion models typically demands a substantial amount of data, we investigate the sensitivity of our method to dataset size. Remarkably, we observe that the performance only experiences only a slight degradation even with significantly smaller sizes. For complete results, please refer to Appendix E.3.

## 5.2 Steering Diffusion Models to Improve their Performance

Having established the effectiveness of relative reward functions in recovering expert goals in the low-dimensional Maze2D environment, we now examine their applicability in higher-dimensional control tasks. Specifically, we evaluate their performance in the HalfCheetah, Hopper, and Walker-2D environments from the D4RL offline locomotion suite [12]. These tasks involve controlling a multi-degree-of-freedom 2D robot to move forward at the highest possible speed. We assess the learned reward functions by examining whether they can enhance the performance of a weak base model when used for classifier-guided steering. If successful, this would provide evidence that the learned relative rewards can indeed bring the base trajectory distribution closer to the expert distribution.

**Implementation.** In these locomotion environments, the notion of reward is primarily focused on moving forward. Therefore, instead of selecting expert behaviors from a diverse set of base behaviors, our reward function aims to guide a below-average-performing base model toward improved performance. Specifically, we train the base models using the `medium-replay` datasets from D4RL, which yield low rewards, and the expert models using the `expert` datasets, which yield high rewards.

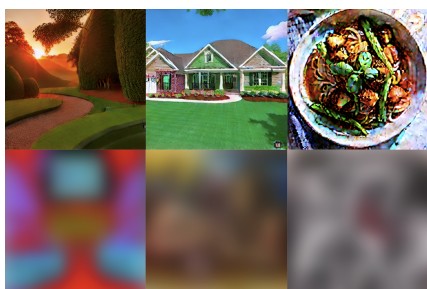

Figure 4: The 3 images with the highest learned reward in their batch ("safe"), and 3 with the lowest reward ("unsafe", blurred), respectively.

Table 1: Diffuser performance with different steering methods, on 3 RL environments and for a low-performance base model (top) and a medium-performance model (bottom).

| Environment | Unsteered | Discriminator | Reward (Ours) |
|---|---|---|---|
| Halfcheetah | $30.38 \pm 0.38$ | $30.41 \pm 0.38$ | $\mathbf{31.65 \pm 0.32}$ |
| Hopper | $24.67 \pm 0.92$ | $25.12 \pm 0.95$ | $\mathbf{27.04 \pm 0.90}$ |
| Walker2d | $28.20 \pm 0.99$ | $27.98 \pm 0.99$ | $\mathbf{38.40 \pm 1.02}$ |
| Mean | 27.75 | 27.84 | **32.36** |
| Halfcheetah | $59.41 \pm 0.87$ | $70.79 \pm 1.92$ | $\mathbf{69.32 \pm 0.8}$ |
| Hopper | $58.8 \pm 1.01$ | $59.42 \pm 2.23$ | $\mathbf{64.97 \pm 1.15}$ |
| Walker2d | $96.12 \pm 0.92$ | $96.75 \pm 1.91$ | $\mathbf{102.05 \pm 1.15}$ |
| Mean | 71.44 | 75.65 | **78.78** |

The reward functions are then fitted using gradient alignment as described in Algorithm 1. Finally, we employ classifier guidance (Section 3) to steer each base model using the corresponding learned reward function.

**Results.** We conduct 512 independent rollouts of the base diffusion model steered by the learned reward, using various guidance scales $\omega$ (Eq. 5). We use the unsteered base model as a baseline. We also compare our approach to a discriminator with the same architecture as our reward function, trained to predict whether a trajectory originates from the base or expert dataset. We train our models with 5 random seeds, and run the 512 independent rollouts for each seed. Steering the base models with our learned reward functions consistently leads to statistically significant performance improvements across all three environments. Notably, the Walker2D task demonstrates a 36.17% relative improvement compared to the unsteered model. This outcome suggests that the reward functions effectively capture the distinctions between the two diffusion models. See Table 1 (top three rows). We further conducted additional experiments in the Locomotion domains in which we steer a *new* medium-performance diffusion model unseen during training, using the relative reward function learned from the base diffusion model and the expert diffusion model. We observed in Table 1 (bottom three rows) that our learned reward function significantly improves performance also in this scenario.

### 5.3 Learning a Reward-Like Function for Stable Diffusion

While reward functions are primarily used in sequential decision-making problems, we propose a generalization to more general domains through the concept of relative reward functions, as discussed in Section 4. To empirically evaluate this generalization, we focus on one of the domains where diffusion models have demonstrated exceptional performance: image generation. Specifically, we examine Stable Diffusion [54], a widely used 859m-parameter diffusion model for general-purpose image generation, and Safe Stable Diffusion [59], a modified version of Stable Diffusion designed to mitigate the generation of inappropriate images.

**Models.** The models under consideration are *latent diffusion models*, where the denoising process occurs in a latent space and is subsequently decoded into an image. These models employ classifier-free guidance [22] during sampling and can be steered using natural language prompts, utilizing CLIP embeddings [49] in the latent space. Specifically, Safe Stable Diffusion [59] introduces modifications to the *sampling loop* of the open-source model proposed by Rombach et al. [54], without altering the model's actual weights. In contrast to traditional classifier-free guidance that steers samples toward a given prompt, the modified sampler of Safe Stable Diffusion also directs samples *away* from undesirable prompts [59, Sec. 3].

**Prompt Dataset.** To investigate whether our reward networks can detect a "relative preference" of Safe Stable Diffusion over the base Stable Diffusion model for harmless images, we use the I2P prompt dataset introduced by Schramowski et al. [59]. This dataset consists of prompts specifically designed to deceive Stable Diffusion into generating imagery with unsafe content. However, we use the dataset to generate sets of *image embeddings* rather than actual images, which serve as training data for our reward networks. A portion of the generated dataset containing an equal number of base and expert samples is set aside for model evaluation.

**Separating Image Distributions.** Despite the complex and multimodal nature of the data distribution in this context, we observe that our reward networks are capable of distinguishing between base and expert images with over 90% accuracy, despite not being explicitly trained for this task. The reward histogram is visualized in Figure 2.

**Qualitative Evaluation.** We find that images that receive high rewards correspond to safe content, while those with low rewards typically contain unsafe or disturbing material, including hateful or offensive imagery. To illustrate this, we sample batches from the validation set, compute the rewards for each image, and decode the image with the highest reward and the one with the lowest reward from each batch. Considering the sensitive nature of the generated images, we blur the latter set as an additional safety precaution. Example images can be observed in Figure 4.

## 6 Conclusion

To the best of our knowledge, our work introduces the first method for extracting relative reward functions from two diffusion models. We provide theoretical justification for our approach and demonstrate its effectiveness in diverse domains and settings. We expect that our method has the potential to facilitate the learning of reward functions from large pre-trained models, improving our understanding and the alignment of the generated outputs. It is important to note that our experiments primarily rely on simulated environments, and further research is required to demonstrate its applicability in real-world scenarios.

## 7 Acknowledgements

This work was supported by the Royal Academy of Engineering (RF\201819\18\163). F.N. receives a scholarship from Fundação Estudar. F.N. also used TPUs granted by the Google TPU Research Cloud (TRC) in the initial exploratory stages of the project.

We would like to thank Luke Melas-Kyriazi, Prof. Varun Kanade, Yizhang Lou, Ruining Li and Eduard Oravkin for proofreading versions of this work.

F.N. would also like to thank Prof. Stefan Kiefer for the support during the project, and Michael Janner for the productive conversations in Berkeley about planning with Diffusion Models.

We use the following technologies and repositories as components in our code: PyTorch [46], NumPy [18], Diffuser [25], D4RL [12], HuggingFace Diffusers [67] and LucidRains's Diffusion Models in Pytorch repository.

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

# Appendix A  More Details on Diffusion Models

## A.1  Diffusion Models in Continuous Time

At a high level, diffusion models work by adding noise to data $\mathbf{x}$, and then learning to denoise it.

The seminal paper on diffusion models [61] and the more recent work of Ho et al. [21] describe this in discrete time. Song et al. [65] show a direct correspondence between the aforementioned formulations of denoising diffusion models and an existing line of work on score-based generative modeling [63], and propose a continuous-time framework leveraging (Itô) Stochastic Differential Equations [36, 45, 58] to unify both methods. We give an overview of the continuous-time formulation, and defer to Appendix A.2 an account of the discrete case.

The forward noising process in continuous time is given by:

$$dx_t = \mathbf{f}(\mathbf{x}_t, t)dt + g(t)d\mathbf{w}_t \tag{15}$$

where $f$ is a function that is Lipschitz, $\mathbf{w}$ is a standard Brownian Motion [36] and $g$ is a "continuous noise schedule" (c.f. A.2). A continuous version of the processes in A.2 [61, 21] is recovered by setting $g(s) = \sqrt{\beta_s}$ and $\mathbf{f}(\mathbf{x}, s) = -\frac{1}{2}\beta_s \mathbf{x}$. The resulting SDE resembles an Ornstein-Uhlenbeck process, which is known to converge in geometric rate to a $\mathcal{N}(0, I)$ distribution. This justifies the choice of distribution of $\mathbf{x}_T$ in A.2.

Song et al. [65] then use a result of Anderson [4] to write the SDE satisfied by the *reverse process* of 2, denoted $\bar{\mathbf{x}}_t$, as:

$$d\bar{\mathbf{x}}_t = \left[\mathbf{f}(\bar{\mathbf{x}}_t, t) - g(t)^2 \nabla_{\mathbf{x}} \log p_t(\bar{\mathbf{x}}_t)\right] dt + g(t)\, d\bar{\mathbf{w}}_t \tag{16}$$

Here $p_t(\mathbf{x})$ denotes the marginal density function of the forward process $\mathbf{x}_t$, and $\bar{\mathbf{w}}$ is a reverse Brownian motion (for further details on the latter, see [4]). The learning step is then to learn an approximation $\mathbf{s}_\Theta(\mathbf{x}_t, t)$ to $\nabla_{\mathbf{x}} \log p_t(\mathbf{x}_t)$ (the *score* of the distribution of $\mathbf{x}_t$). This is shown by Song et al. [65] to be equivalent to the noise model $\boldsymbol{\epsilon}_\Theta$ in A.2. The sampling step consists of taking $\bar{\mathbf{x}}_T \sim \mathcal{N}(0, I)$ for some $T > 0$, and simulating 16 *backwards in time* to arrive at $\bar{\mathbf{x}}_0 \approx \mathbf{x}_0$.

One of the advantages of this perspective is that the sampling loop can be offloaded to standard SDE solvers, which can more flexibly trade-off e.g. computation time for performance, compared to hand-designed discretizations [32].

## A.2  Discrete-Time Diffusion Models

The seminal paper on diffusion models [61] and the more recent work of Ho et al. [21] formulate diffusion models in discrete time. The forward noising is a Markov Chain starting from an uncorrupted data point $\mathbf{x}_0$, and repeatedly applying a (variance-preserving) Gaussian kernel to the data:

$$q_t(\mathbf{x}_{t+1}|\mathbf{x}_t) \sim \mathcal{N}(\sqrt{1-\beta_t}\mathbf{x}_t, \beta_t I)$$

The coefficients $\beta_t$ are known as the *noise schedule* of the diffusion model, and can be chosen to regulate the speed with which noise is added to the data [32].

The modeling step then aims to *revert* the noising process by learning a backward kernel

$$p_\theta(\mathbf{x}_{t-1}|\mathbf{x}_t) \sim \mathcal{N}(\boldsymbol{\mu}_\theta(\mathbf{x}_t, t), \sigma_t^2) \tag{17}$$

The seminal paper of Sohl-Dickstein et al. [61] proposes a variational loss for training $\mu_\theta$, based on a maximum-likelihood framework. Subsequently, Ho et al. [21] proposed a new parametrization of $\mu_\theta$, as well as a much simpler loss function. Their loss function is

$$L_{\text{simple}}(\theta) := \mathbb{E}_{t, \mathbf{x}_0, \boldsymbol{\epsilon}}\left[\left\|\boldsymbol{\epsilon} - \boldsymbol{\epsilon}_\theta(\sqrt{\bar{\alpha}_t}\mathbf{x}_0 + \sqrt{1-\bar{\alpha}_t}\boldsymbol{\epsilon}, t)\right\|^2\right] \tag{18}$$

where $\alpha_t := 1 - \beta_t$, $\bar{\alpha}_t := \prod_{s=1}^{t} \alpha_s$ and $\boldsymbol{\mu}_\theta(\mathbf{x}_t, t) = \frac{1}{\sqrt{\alpha_t}}\left(\mathbf{x}_t - \frac{\beta_t}{\sqrt{1-\bar{\alpha}_t}}\boldsymbol{\epsilon}_\theta(\mathbf{x}_t, t)\right)$. Intuitively, this loss says that $\boldsymbol{\epsilon}_\theta(\mathbf{x}_t, t)$ should fit the noise $\boldsymbol{\epsilon}$ added to the initial datapoint $\mathbf{x}_0$ up to time $t$. For a full account of this derivation, we refer the reader to Ho et al. [21].

### A.2.1  Classifier Guidance

Sohl-Dickstein et al. [61] also propose a method for *steering* the sampling process of diffusion models. Intuitively, if the original diffusion model computes a distribution $p(\mathbf{x}_0)$, we would like to make it compute $p(\mathbf{x}_0|y)$, where we assume, for the sake of concreteness, that $y \in \{0, 1\}$. Applying Bayes's rule gives the factorization $p(\mathbf{x}_0|y) \propto p(\mathbf{x}_0) \cdot p(y|\mathbf{x}_0)$. The first term is the original distribution the diffusion model is trained on. The second term is a probabilistic classifier, inferring the probability that $y = 1$ for a given sample $\mathbf{x}_0$.

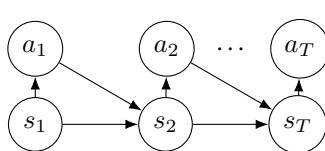

(a) Probabilistic Graphical Model (PGM) for sampling from a Markov Decision Process (MDP) with a fixed policy $\pi$.

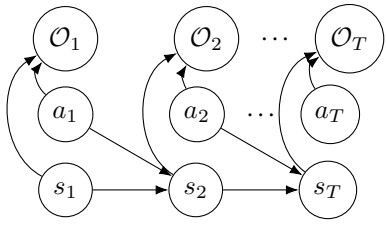

(b) PGM for planning, with optimality variables $\mathcal{O}_{1:T}$.

Figure 5: Reinforcement Learning as Probabilistic Inference (see App. B for a detailed explanation).

We now leverage the framework in A.1 to derive the classifier guidance procedure. Equation 16 shows we can sample from the backward process by modeling $\nabla_{\mathbf{x}}\log p_t(\mathbf{x}_t)$ using $\mathbf{s}_\Theta(\mathbf{x}_t, t)$. Thus, to model $p(\mathbf{x}_0|y)$, we may instead approximate $\nabla_{\mathbf{x}}\log p(\mathbf{x}_t|y)$ using:

$$\nabla_{\mathbf{x}}\log p(\mathbf{x}_t|y) = \nabla_{\mathbf{x}}\log \frac{p(\mathbf{x}_t)\, p(y|\mathbf{x}_t)}{p(y)} \tag{19}$$

$$= \nabla_{\mathbf{x}}\log p(\mathbf{x}_t) + \nabla_{\mathbf{x}}\log p(y|\mathbf{x}_t) \tag{20}$$

$$\approx \mathbf{s}_\Theta(\mathbf{x}_t, t) + \nabla_{\mathbf{x}}\rho(\mathbf{x}_t, t) \tag{21}$$

where $\rho(\mathbf{x}, t)$ is a neural network approximating $\log p(y|\mathbf{x}_t)$. In practice, we multiply the gradients of $\rho$ by a small constant $\omega$, called the *guidance scale*.

The *guided* reverse SDE hence becomes:

$$d\bar{\mathbf{x}}_t = \left[\mathbf{f}(\bar{\mathbf{x}}_t, t) - g(t)^2[\mathbf{s}_\Theta(\bar{\mathbf{x}}_t, t) + \omega\nabla_{\mathbf{x}}\rho(\bar{\mathbf{x}}_t, t)]\right]dt + g(t)\,d\bar{\mathbf{w}}_t \tag{22}$$

The main takeaway is, informally, that we can steer a diffusion model to produce samples with some property $y$ by gradually pushing the samples in the direction that maximizes the output of a classifier predicting $p(y|\mathbf{x})$.

## Appendix B    More Details on Reinforcement Learning as Probabilistic Inference

Levine [37] provides an in-depth exposition on existing methods for approaching sequential decision-making from a probabilistic and causal angle, referencing the work of e.g. [66, 70, 71, 31, 30]. We refer to it as the Control as Inference Framework (CIF).

The starting point for the formulation is the classical notion of a Markov Decision Problem (MDP), a stochastic process given by a state space $\mathcal{S}$, an action space $\mathcal{A}$, a (possibly stochastic) transition function $\mathcal{T} : \mathcal{S} \times \mathcal{A} \to \Delta_{\mathcal{S}}$, and a reward function $r : \mathcal{S} \times \mathcal{A} \to \Delta_{\mathbb{R}_{\geq 0}}$. Here $\Delta_{\mathcal{X}}$ denotes the set of distributions over a set $\mathcal{X}$. In this context, a policy $\pi : \mathcal{S} \to \Delta_{\mathcal{A}}$ represents a (possibly stochastic) way of choosing an action given the current state.

The MDP starts at an initial state $s_0 \in \mathcal{S}$, and evolves by sampling $a_t \sim \pi(s_t)$, and then $s_{t+1} \sim \mathcal{T}(s_t, a_t)$ for $t \geq 0$. The reward received at time $t$ is $r_t \sim r(s_t, a_t)$. We consider here an episodic setting, where the MDP stops at a fixed time $T > 0$. We call the sequence $((s_t, a_t))_{t=0}^T$ a *trajectory*. Sampling corresponds to the Probabilistic Graphical Model (PGM) in 5a, where each action depends on the current state, and the next state depends on the current state and the current action.

To encode the process of *choosing a policy to optimize a reward*, this PGM can be extended with *optimality variables*, as per Fig. 5b. They are defined as $\mathcal{O}_t \sim \text{Ber}(e^{-r(s_t, a_t)})$, so that their *distribution* encodes the dependency of reward on current states and actions. The problem of optimizing the reward can then be recast as *sampling from* $p(\tau|\mathcal{O}_{1:T})$. Following [37], one can apply Bayes's Rule to obtain

$$p(\tau|\mathcal{O}_{1:T}) \propto p(\tau) \cdot p(\mathcal{O}_{1:T}|\tau) \tag{23}$$

which factorizes the distribution of optimal trajectories (up to a normalizing constant) as a *prior $p(\tau)$* over trajectories and a *likelihood term $p(\mathcal{O}_{1:T}|\tau)$*. Observe that, from the definition of $\mathcal{O}_{1:T}$ and the PGM factorization in Fig. 5b, we have $p(\mathcal{O}_{1:T}|\tau) = e^{-\sum_t r(s_t, a_t)}$. Hence, in the CIF, $-\log p(\mathcal{O}_{1:T}|\tau)$ (a *negative log likelihood*) corresponds to the cumulative reward of a trajectory $\tau$.

# Appendix C  Details on Main Derivations

## C.1  Existence and Uniqueness of Strong Solutions to Stochastic Differential Equations

**Theorem C.1** (Existence and uniqueness theorem for SDEs, c.f. page 66 of Øksendal and Øksendal [45]). *Let $T > 0$ and $b(\cdot, \cdot) : [0, T] \times \mathbb{R}^n \to \mathbb{R}^n, \sigma(\cdot, \cdot) : [0, T] \times \mathbb{R}^n \to \mathbb{R}^{n \times m}$ be measurable functions satisfying*

$$|b(t, x)| + |\sigma(t, x)| \le C(1 + |x|); \quad x \in \mathbb{R}^n, t \in [0, T]$$

*for some constant $C$, where $|\sigma|^2 = \sum |\sigma_{ij}|^2$, and such that*

$$|b(t, x) - b(t, y)| + |\sigma(t, x) - \sigma(t, y)| \le D|x - y|; \quad x, y \in \mathbb{R}^n, t \in [0, T]$$

*for some constant $D$. Let $Z$ be a random variable which is independent of the $\sigma$-algebra $\mathcal{F}_\infty^{(m)}$ generated by $B_s(\cdot), s \ge 0$ and such that*

$$\mathbb{E}\left[|Z|^2\right] < \infty$$

*Then the stochastic differential equation*

$$dX_t = b(t, X_t) \, dt + \sigma(t, X_t) \, dB_t, \quad 0 \le t \le T, X_0 = Z$$

*has a unique $t$-continuous solution $X_t(\omega)$ with the property that $X_t(\omega)$ is adapted to the filtration $\mathcal{F}_t^Z$ generated by $Z$ and $B_s(\cdot); s \le t$ and*

$$\mathbb{E}\left[\int_0^T |X_t|^2 \, dt\right] < \infty.$$

*Remark* C.2. In the above, the symbol $|\cdot|$ is overloaded, and taken to mean the *norm* of its argument. As everything in sight is finite-dimensional, and as all norms in finite-dimensional normed vector spaces are topologically equivalent (Theorem 2.4-5 in Kreyszig [34]), the stated growth conditions do not depend on the particular norm chosen for $\mathbb{R}^n$.

Also, whenever we talk about a *solution to an SDE* with respect to e.g. the standard Brownian motion $\mathbf{w}$ and an initial condition $\mathbf{z}$, we are always referring only to $(\mathcal{F}_t^{\mathbf{z}})_{t \ge 0}$-adapted stochastic processes, as per Theorem C.1.

**Corollary C.3.** *Fix a Brownian Motion $\mathbf{w}$ and let $(\mathcal{F}_t)_{t \ge 0}$ be its natural filtration. Consider an SDE $d\mathbf{x}_t = \mathbf{f}(\mathbf{x}_t, t) + g(t) d\mathbf{w}_t$ with initial condition $\mathbf{z} \in \mathcal{L}^2$ independent of $\mathcal{F}_\infty = \cup_{t \ge 0} \mathcal{F}_t$. Suppose $\mathbf{f} : \mathbb{R}^n \times [0, T] \to \mathbb{R}^n$ is Lipschitz with respect to $(\mathbb{R}^n \times [0, T], \mathbb{R}^n)$ and $g : [0, T] \to \mathbb{R}_{\ge 0}$ is continuous and bounded.*

*Then the conclusion of C.1 holds and there is an a.s. unique $(\mathcal{F}_t^{\mathbf{z}})_{t \ge 0}$-adapted solution $(\mathbf{x}_t)_{t \ge 0}$ having a.s. continuous paths, where $(\mathcal{F}_t^{\mathbf{z}})_{t \ge 0}$ is defined as in C.1.*

*Proof.* Firstly, note that, as $\mathbf{f}$ and $g$ are continuous, they are Lebesgue-measurable. It remains to check the growth conditions in C.1.

As $\mathbf{f}$ is Lipschitz, there exists $C_1$ (w.l.o.g. $C_1 > 1$) such that, for any $\mathbf{x}, \mathbf{y} \in \mathbb{R}^n$ and $t, s \in [0, T]$:

$$|\mathbf{f}(\mathbf{x}, t) - \mathbf{f}(\mathbf{y}, s)| \le C_1(|\mathbf{x} - \mathbf{y}| + |t - s|) \tag{24}$$

$$\le C_1(|\mathbf{x} - \mathbf{y}| + T) \tag{25}$$

In particular, taking $\mathbf{y} = \mathbf{0}$ and $s = 0$, we obtain that $|\mathbf{f}(\mathbf{x}, t) - \mathbf{f}(\mathbf{0}, 0)| \le C_1(|\mathbf{x}| + T)$. Let $M = |\mathbf{f}(\mathbf{0}, 0)|/C_1$. Then, by the triangle inequality, $|\mathbf{f}(\mathbf{x}, t)| \le C_1(|\mathbf{x}| + T + M)$.

As $g$ is bounded, there exists $C_2 > 0$ such that $|g(t)| \le C_2$, and so for any $t, s \in [0, T]$, we have:

$$|g(t) - g(s)| \le 2C_2 \tag{26}$$

Hence we have, for $t \in [0, T]$ and $\mathbf{x}, \mathbf{y} \in \mathbb{R}^n$:

$$|\mathbf{f}(\mathbf{x}, t)| + |g(t)| \le ((T + M)C_1 + C_2)(1 + |\mathbf{x}|) \qquad |\mathbf{f}(\mathbf{x}, t) - \mathbf{f}(\mathbf{y}, t)| \le C_1|\mathbf{x} - \mathbf{y}| \tag{27}$$

which yields the growth conditions required in C.1, and the conclusion follows. $\square$

## C.2  Proof of Theorem 4.1

*Proof of Theorem 4.1.* We reference the fundamental result on the existence and uniqueness of strong solutions to SDEs, presented in Section 5.2.1 of [45] and reproduced above. In particular, Corollary C.3 shows that the Existence and Uniqueness Theorem C.1 applies for SDEs 6 and 7, by our assumptions on $\mathbf{f}^{(1)}$, $\mathbf{f}^{(2)}$ and $g$. As also we restrict our attention to $\mathbf{h}$ that are Lipschitz, and sums of Lipschitz functions are also Lipschitz, the corollary also applies to SDE 8. This establishes the existence of a.s. unique (adapted) solutions with a.s. continuous sample paths for these SDEs. Call them $\mathbf{x}^{(1)}$, $\mathbf{x}^{(2)}$ and $\mathbf{x}$, respectively.

Denote $\mathbf{h}_1(\mathbf{x}, t) = \mathbf{f}^{(2)}(\mathbf{x}, t) - \mathbf{f}^{(1)}(\mathbf{x}, t)$. The choice $\mathbf{h} = \mathbf{h}_1$ makes SDE 8 have the same drift and noise coefficients as SDE 7, and so, by the uniqueness of solutions we established, it follows that $\mathbf{x}_t^{(2)} = \mathbf{x}_t$ for all $t$ a.s., with this choice of $\mathbf{h}$.

Now suppose we have a Lipschitz-continuous $\tilde{\mathbf{h}}$ which yields an a.s. unique, $t$-continuous solution $(\tilde{\mathbf{x}}_t)_{t \geq 0}$ that is indistinguishable from $\mathbf{x}^{(2)}$. (i.e. equal to $\mathbf{x}^{(2)}$ for all $t$, a.s.). We show that $\tilde{\mathbf{h}} = \mathbf{h}_1$.

As $\tilde{\mathbf{x}}$ and $\mathbf{x}^{(2)}$ are indistinguishable and satisfy SDEs with the same noise coefficient, we obtain a.s., for any $t \in [0, T]$:

$$0 = \tilde{\mathbf{x}}_t - \mathbf{x}_t^{(2)} \tag{28}$$

$$= \left( \int_0^t \mathbf{f}^{(1)}(\tilde{\mathbf{x}}_s, s) + \tilde{\mathbf{h}}(\tilde{\mathbf{x}}_s, s) ds + \int_0^t g(s) \, d\mathbf{w}_s \right) - \left( \int_0^t \mathbf{f}^{(2)}(\mathbf{x}_s^{(2)}, s) ds + \int_0^t g(s) \, d\mathbf{w}_s \right) \tag{29}$$

$$= \int_0^t \mathbf{f}^{(1)}(\tilde{\mathbf{x}}_s, s) + \tilde{\mathbf{h}}(\tilde{\mathbf{x}}_s, s) ds - \int_0^t \mathbf{f}^{(2)}(\mathbf{x}_s^{(2)}, s) ds \tag{30}$$

$$= \int_0^t \mathbf{f}^{(1)}(\mathbf{x}_s^{(2)}, s) + \tilde{\mathbf{h}}(\mathbf{x}_s^{(2)}, s) ds - \int_0^t \mathbf{f}^{(2)}(\mathbf{x}_s^{(2)}, s) ds \tag{31}$$

$$= \int_0^t \tilde{\mathbf{h}}(\mathbf{x}_s^{(2)}, s) - (\mathbf{f}^{(2)}(\mathbf{x}_s^{(2)}, s) - \mathbf{f}^{(1)}(\mathbf{x}_s^{(2)}, s)) ds \tag{32}$$

$$= \int_0^t \tilde{\mathbf{h}}(\mathbf{x}_s^{(2)}, s) - \mathbf{h}_1(\mathbf{x}_s^{(2)}, s) ds \tag{33}$$

where in 29 we substitute the integral forms of the SDEs satisfied by $\tilde{\mathbf{x}}$ and $\mathbf{x}^{(2)}$, and in 31 we use that the processes are indistinguishable to replace $\tilde{\mathbf{x}}$ by $\mathbf{x}^{(2)}$ in one of the integrals.

Hence, with probability 1, $\int_0^t \tilde{\mathbf{h}}(\mathbf{x}_s^{(2)}, s) ds = \int_0^t \mathbf{h}_1(\mathbf{x}_s^{(2)}, s) ds$. As $\mathbf{x}^{(2)}$ is a.s. continuous, we also have that the mappings $s \mapsto \mathbf{h}_1(\mathbf{x}_s^{(2)}, s)$ and $s \mapsto \tilde{\mathbf{h}}(\mathbf{x}_s^{(2)}, s)$ are continuous with probability 1. We may hence apply the Fundamental Theorem of Calculus and differentiate 33 to conclude that a.s. $\tilde{\mathbf{h}}(\mathbf{x}_s^{(2)}, s) = \mathbf{h}_1(\mathbf{x}_s^{(2)}, s)$ for all $s$.

As $g(0) > 0$, $\mathbf{x}_t^{(2)}$ is supported on all of $\mathbb{R}^n$ for any $t > 0$. Therefore, the above implies that $\tilde{\mathbf{h}}(\mathbf{x}, s) = \mathbf{h}_1(\mathbf{x}, s)$ for all $s \in (0, T]$ *and* all $\mathbf{x} \in \mathbb{R}^n$.

More formally, let $\Delta(\mathbf{x}', s) = \tilde{\mathbf{h}}(\mathbf{x}', s) - \mathbf{h}_1(\mathbf{x}', s)$. Then $\Delta(\mathbf{x}_s^{(2)}, s) = 0$ for all $s$ a.s.. Also, $\Delta$ is Lipschitz. Let $C$ be its Lipschitz constant. Take $\mathbf{x} \in \mathbb{R}^n$ and fix $\varepsilon > 0$ and $t > 0$. Then $\mathbb{P}(||\mathbf{x}_t^{(2)} - \mathbf{x}||_2 < \frac{\varepsilon}{2C}) > 0$, as $\mathbf{x}_t^{(2)}$ is supported in all of $\mathbb{R}^n$. Hence, by the above, there exists $\mathbf{y}_\varepsilon \in B(\mathbf{x}, \frac{\varepsilon}{2C})$ such that $\Delta(\mathbf{y}_\varepsilon, t) = 0$. As $\tilde{\mathbf{h}}$ is Lipschitz, for any $\mathbf{x}' \in B(\mathbf{x}, \frac{\varepsilon}{2C})$, we have that

$$||\Delta(\mathbf{x}', t)|| \leq C||\mathbf{x}' - \mathbf{y}_\varepsilon|| \tag{34}$$

$$\leq C(||\mathbf{x}' - \mathbf{x}|| + ||\mathbf{x} - \mathbf{y}_\varepsilon||) \tag{35}$$

$$\leq C(\frac{\varepsilon}{2C} + \frac{\varepsilon}{2C}) \tag{36}$$

$$= \varepsilon \tag{37}$$

As $\varepsilon$ was arbitrary, we have that $\Delta(\mathbf{x}', t) \to 0$ as $\mathbf{x}' \to \mathbf{x}$. As $\Delta$ is continuous (since it is Lipschitz), it follows that $\Delta(\mathbf{x}, t) = 0$. As $\mathbf{x}$ was also arbitrary, we have that $\tilde{\mathbf{h}}(\mathbf{x}, s) = \mathbf{h}_1(\mathbf{x}, s)$ for all $s \in (0, T]$ *and* all $\mathbf{x} \in \mathbb{R}^n$.

As $\mathbf{h}$ must be chosen to be (Lipschitz) continuous also with respect to $t$, for any $\mathbf{x}$ it must be that

$$\tilde{\mathbf{h}}(\mathbf{x}, 0) = \lim_{s \to 0^+} \tilde{\mathbf{h}}(\mathbf{x}, s) = \lim_{s \to 0^+} \mathbf{h}(\mathbf{x}, s) = \mathbf{h}_1(\mathbf{x}, 0) \tag{38}$$

This completes the proof of the uniqueness of the choice $\mathbf{h} = \mathbf{h}_1$. $\qquad \square$

## C.3 Does the Backward Process of a Diffusion Model Fit in Theorem C.1?

In the forward process 2, $\mathbf{f}$ is taken to be Lipschitz, and $g$ is always bounded, e.g. in [21, 62, 65]. However, in the backward process, one may be concerned about whether the score term $-\nabla_{\mathbf{x}} \log p_t(\mathbf{x}_t, t)$ is Lipschitz. We argue that it is reasonable to assume that it is.

We consider a setting where $g(t) = 1$, and $\mathbf{f}(\mathbf{x}, t) = -\lambda \mathbf{x}$. Then the forward process is the well-understood Ornstein-Uhlenbeck (or Langevin) process [45, p. 74]:

$$d\mathbf{x}_t = -\lambda \mathbf{x}_t \, dt + d\mathbf{w}_t \tag{39}$$

Consider the simplified case where the initial condition is deterministic: $\mathbf{x}_0 = \mathbf{y} \in \mathbb{R}^n$. Then $\mathbf{x}$ is a Gaussian process of co-variance function given by $K(s,t) = \text{cov}(X_s, X_t) = \frac{e^{-\lambda|t-s|} - e^{-\lambda(t+s)}}{2\lambda} I$, where $I$ is the identity matrix, as per Le Gall [36], page 226. Its mean function is $\mathbf{m}_t = \mathbf{y}e^{-\lambda t}$. In particular, it has variance $\sigma_t^2 = \frac{1 - e^{-2\lambda t}}{2\lambda}$, and $\sigma_t^2 \to \frac{1}{2\lambda}$ as $t \to \infty$.

We follow the theoretical analysis of the convergence of diffusion models by De Bortoli [9] and assume the backward diffusion process finishes at a time $\varepsilon > 0$. Hence, the backward process $\bar{\mathbf{x}}_t$ is now supported in all of $\mathbb{R}^n$ for any $t \in [\epsilon, T]$. The original data need not be supported everywhere, and in fact is often assumed to not be, c.f. the manifold hypothesis in [9]. In our simplified case, it is supported at a point. This is why $t = 0$ needs to be excluded by the backward process finishing at some $\varepsilon > 0$.

From the aforementioned properties of the Ornstein-Uhlenbeck process, $p_t(\mathbf{x}_t) = C \exp\left(-\frac{||\mathbf{x}_t - \mathbf{m}_t||_2^2}{2\sigma_t^2}\right)$, where $C$ is a constant independent of $\mathbf{x}$. But then $-\nabla_\mathbf{x} \log p_t(\mathbf{x}_t, t) = \frac{\mathbf{x}_t - \mathbf{m}_t}{\sigma_t^2}$. For any fixed $t$, we hence have that $-\nabla_\mathbf{x} \log p_t(\mathbf{x}_t, t)$ is $1/\sigma_t^2$-Lipschitz. Since the variance $\sigma_t$ is bounded away from 0 for $t \in [\varepsilon, T]$, we may pick a Lipschitz constant that works uniformly across all $\mathbf{x}$ and $t$ (for instance, $1/\sigma_\varepsilon^2$).

This can be extended to the case where the initial data is bounded, which is frequently assumed (e.g. by De Bortoli [9]). Informally, if the data is bounded, it cannot change the *tails* of the distribution of $\mathbf{x}_t$ significantly. We omit further details, as this discussion is beyond the scope of the paper.

In summary, the forward noising process is very similar to an Ornstein-Uhlenbeck process, so, for practical purposes, it is reasonable to assume the scores $\nabla_\mathbf{x} \log p_t(\mathbf{x}_t, t)$ are Lipschitz.

## C.4   Existence and Uniqueness of Minimizer of (11)

**Definition C.4.** $L^2(\mathbb{R}^n, \mathbb{R}^n)$ is the Hilbert space given by

$$L^2(\mathbb{R}^n, \mathbb{R}^n) = \left\{ \mathbf{f} : \mathbb{R}^n \to \mathbb{R}^n : \int_{\mathbb{R}^n} ||\mathbf{f}(\mathbf{x})||_2^2 d\mathbf{x} < \infty \right\} \tag{40}$$

*Remark* C.5. It is easy to show $L^2(\mathbb{R}^n, \mathbb{R}^n)$ is a Hilbert Space with norm given by

$$||\mathbf{f}||_2^2 = \int_{\mathbb{R}^n} ||\mathbf{f}(\mathbf{x})||_2^2 \, d\mathbf{x} \tag{41}$$

For instance, one can start from the standard result that $L^2(\mathbb{R}^n, \mathbb{R})$ is a Hilbert Space, and apply it to each coordinate of the output of $\mathbf{f}$.

**Definition C.6.** Denote by $\text{Cons}(\mathbb{R}^n)$ the space of the gradients of smooth, square-integrable potentials from $\mathbb{R}^n$ to $\mathbb{R}$ with square-integrable gradient, i.e.

$$\text{Cons}(\mathbb{R}^n) = \left\{ \nabla f : f \text{ is smooth}, \int_{\mathbb{R}^n} |f(\mathbf{x})|^2 d\mathbf{x} < \infty \text{ and } \int_{\mathbb{R}^n} ||\nabla f(\mathbf{x})||_2^2 d\mathbf{x} < \infty \right\} \tag{42}$$

*Remark* C.7. Clearly $\text{Cons}(\mathbb{R}^n) \subseteq L^2(\mathbb{R}^n, \mathbb{R}^n)$. The condition that "$f$ is square-integrable and has a square-integrable gradient" corresponds to $f$ being in the Sobolev space $W^{1,2}(\mathbb{R}^n)$ (see Jost [29], Chapter 9).

Denote by $\overline{\text{Cons}(\mathbb{R}^n)}$ the **closure** of $\text{Cons}(\mathbb{R}^n)$, i.e.

$$\overline{\text{Cons}(\mathbb{R}^n)} = \{\mathbf{f} : \text{there exists } (\nabla f_k)_{k \geq 0} \subseteq \text{Cons}(\mathbb{R}^n) \text{ such that } ||\nabla f_k - \mathbf{f}||_2^2 \to 0 \text{ as } k \to \infty\} \tag{43}$$

By construction, $\overline{\text{Cons}(\mathbb{R}^n)}$ is a vector subspace of $L^2(\mathbb{R}^n, \mathbb{R}^n)$, and is closed, i.e. stable under taking limits.

**Theorem C.8** (Complementation in Hilbert Spaces, c.f. Kreyszig [34], Theorem 3.3-4)**.** *Let* $(\mathcal{H}, \langle \cdot, \cdot \rangle, ||\cdot||)$ *be a Hilbert space, and let* $Y \subseteq H$ *be a closed vector subspace. Then any* $x \in \mathcal{H}$ *can be written as* $x = y + z$, *where* $y \in Y$ *and* $z \in Y^\perp$.

**Corollary C.9** (Uniqueness of Projection)**.** *Then the minimum of* $v \mapsto ||v - x||$ *over* $v \in Y$ *is attained at the (unique)* $y$ *given in the theorem, as* $||v - x||^2 \geq |\langle v - x, z \rangle| = ||z||^2$, *and setting* $v = y$ *attains this bound.*

*Proof of Proposition 4.3.* Firstly note that we assumed $\mathbf{s}_\Theta^{(2)}(\cdot, t) - \mathbf{s}_\phi^{(1)}(\cdot, t)$ is in $L^2(\mathbb{R}^n, \mathbb{R}^n)$ for each individual $t$.

It follows directly from the above that, as $\overline{\text{Cons}(\mathbb{R}^n)}$ is a closed subspace of $L^2(\mathbb{R}^n, \mathbb{R}^n)$, Corollary C.9 applies, and we have that there is a unique minimizer $\mathbf{h}_t \in \overline{\text{Cons}(\mathbb{R}^n)}$ in Equation 11.

As $\overline{\text{Cons}(\mathbb{R}^n)}$ consists of $L^2(\mathbb{R}^n, \mathbb{R}^n)$ limits of sequences in $\text{Cons}(\mathbb{R}^n)$, there exists a sequence $(\nabla \Phi_k)_{k \geq 0}$ of gradients of $W^{1,2}$-potentials such that $||\nabla \Phi_k - \mathbf{h}_t||_2^2 \to 0$ as $k \to \infty$. From the definition of convergence, we

get that, for any $\varepsilon > 0$, there is $k$ large enough such that

$$\int_{\mathbb{R}^n} ||\nabla_{\mathbf{x}} \Phi_k(\mathbf{x}) - \mathbf{h}_t(\mathbf{x})||_2^2 \, d\mathbf{x} < \varepsilon \tag{44}$$

which completes the proof. $\qquad\square$

## Appendix D   Relative Reward Learning Algorithms

The below algorithm was used for learning the relative reward functions for experiments with Stable Diffusion (see Section 5.3).

---
**Algorithm 2:** Relative reward function training with access only to diffusion models

---
**Input:** Base diffusion model $\mathbf{s}^{(1)}$ and expert diffusion model $\mathbf{s}^{(2)}$.
**Output:** Relative reward estimator $\rho_\theta$.
// Dataset pre-generation using the diffusion models
$\mathcal{D}_1 \leftarrow \emptyset$
$\mathcal{D}_2 \leftarrow \emptyset$
**for** $m \in \{1, 2\}$, $K$ *times* **do**
$\quad$ $\mathbf{x}_T \sim \mathcal{N}(0, I)$
$\quad$ **for** $t \in T - 1, ..., 0$ **do**
$\quad\quad$ $\mathbf{x}_t^{(1)} \leftarrow \mathbf{x}_{t+1}$ denoised by 1 more step using $\mathbf{s}^{(1)}$
$\quad\quad$ $\mathbf{x}_t^{(2)} \leftarrow \mathbf{x}_{t+1}$ denoised by 1 more step using $\mathbf{s}^{(2)}$
$\quad\quad$ $\mathbf{x}_t \leftarrow \mathbf{x}_t^{(m)}$
$\quad\quad$ Add $(t + 1, \mathbf{x}_{t+1}, \mathbf{x}_t^{(1)}, \mathbf{x}_t^{(2)})$ to $\mathcal{D}_m$
$\quad$ **end**
**end**

// Training
$\mathcal{D} \leftarrow \mathcal{D}_1 \cup \mathcal{D}_2$
Initialize parameters $\theta$
**for** $i \in 1...n\_train\_steps$ **do**
$\quad$ // Using batch size of 1 for clarity
$\quad$ Sample $(t + 1, \mathbf{x}_{t+1}, \mathbf{x}_t^{(1)}, \mathbf{x}_t^{(2)})$ from $\mathcal{D}$
$\quad$ // use pre-computed diffusion outputs to compute loss
$\quad$ $\hat{L}_{\text{RRF}}(\theta) \leftarrow ||\nabla_{\mathbf{x}} \rho_\theta(\mathbf{x}_{t+1}, t + 1) - (\mathbf{x}_t^{(2)} - \mathbf{x}_t^{(1)})||_2^2$
$\quad$ Take an Adam [33] optimization step on $\theta$ according to $\hat{L}_{\text{IRL}}(\theta)$
**end**

---

*Remark* D.1. In our experiments with Stable Diffusion, the generation of the datasets $\mathcal{D}_1$ and $\mathcal{D}_2$ uses a *prompt dataset* for generating the images. For each prompt, we generate one image following the base model ($m = 1$), and one following the expert model ($m = 2$). The I2P prompt dataset [59] contains 4703 prompts.

## Appendix E   Implementation Details, Additional Results and Ablations

Table 2: Reward achieved by the agent when trained either with the groundtruth reward or with the extractive relative reward (Ours). Note that a policy taking random actions achieves close to zero reward.

| Environment | Groundtruth Reward | Relative Reward (Ours) |
|---|---|---|
| OpenMaze | $92.89 \pm 11.79$ | $76.45 \pm 19.10$ |
| UMaze | $94.94 \pm 8.89$ | $74.52 \pm 18.32$ |
| MediumMaze | $423.21 \pm 51.30$ | $276.10 \pm 65.21$ |
| LargeMaze | $388.76 \pm 121.39$ | $267.56 \pm 98.45$ |

Table 3: Performance bounds for ablation of t-stopgrad and guide scales.

| Environment | Lower Bound | Mean | Upper Bound |
|---|---|---|---|
| HalfCheetah | $30.24 \pm 0.39$ | 30.62 | $31.5 \pm 0.35$ |
| Hopper | $19.65 \pm 0.27$ | 22.32 | $25.03 \pm 0.65$ |
| Walker2d | $31.65 \pm 0.92$ | 34.35 | $38.14 \pm 1.08$ |

## E.1 Implementation Details

### E.1.1 Diffusion Models.

Following [25] Section 3, we use the DDPM framework of Ho et al. 2022 with a cosine beta schedule. We do not use preconditioning in the sense of Section 5 of [32]. However, we do clip the the denoised samples during sampling, and apply scaling to trajectories, as discussed below in Appendix E.1.6.

### E.1.2 Relative Reward Function Architectures.

The architectures and dimensions for all experiments are given in Table 4. In the Locomotion environments and in Stable Diffusion, we parameterize the relative reward functions using the encoder of a UNet, following the architecture of Janner et al. [25] for parameterizing their value function networks. In the Maze environments, we use an MLP taking in a sequence of $h$ state-action pairs. In the following, the *dimensions* refer to the number of channels after each downsampling step in the case of UNet architectures, and to layer dimensions in the case of MLP architectures. The *horizon* refers to the number of state-action pairs input to the reward function network. When it does not match the diffusion horizon $H$, we split trajectories into $H/h$ segments, feed each into the reward network, and average the results.

### E.1.3 Relative Reward Function Training Hyperparameters.

In Table 5 we indicate the learning rate and batch size used for training the reward functions, as well as the number of denoising timesteps of the diffusion models they are trained against. We report the number of training steps for the models used to generate the plots and numerical results in the main paper. We use Adam [46] as an optimizer, without weight decay. The Diffusion models use ancestral sampling as in [25, 21], with a cosine noise schedule [44].

### E.1.4 Conditional Generation Hyperparameters.

In Locomotion environments, we find that the performance of the steered model is affected by the guidance scale $\omega$ and by `t_stopgrad`, a hyperparameter specifying the step after which we *stop* adding reward gradients to the diffusion model sample. For example, if `t_stopgrad = 5`, we only steer the generation up to timestep 5, and for steps 4 down to 0 no more steering is used.

In Tables 6 and 7 we indicate the respective values used for the results reported in Table 1 (see *Discriminator* and *Reward (Ours)*), as well as for the results presented in the later presented ablation study in Appendix E.3.2 (see *Reward (Ours, Ablation)*).

### E.1.5 Using Estimates of $\mathbb{E}[\mathbf{x}_{t-1}|\mathbf{x}_0, \mathbf{x}_t]$ for Gradient Alignment.

Diffusion models can be trained to output score estimates $\mathbf{s}_\theta(x_t, t) \approx \nabla_{\mathbf{x}}\log p_t(\mathbf{x}_t)$, but also estimates of a denoised sample: $\boldsymbol{\mu}_\theta(\mathbf{x}_t, t) = \frac{1}{\sqrt{\alpha_t}}\left(\mathbf{x}_t - \frac{\beta_t}{\sqrt{1-\bar{\alpha}_t}}\mathbf{s}_\theta(\mathbf{x}_t, t)\right) \approx \mathbb{E}[\mathbf{x}_{t-1}|\mathbf{x}_0, \mathbf{x}_t]$.

For two diffusion models $\mathbf{s}_\phi^{(1)}$ and $\mathbf{s}_\Theta^{(2)}$, we have that $\boldsymbol{\mu}_\phi(\mathbf{x}_t, t) - \boldsymbol{\mu}_\Theta(\mathbf{x}_t, t) \propto \mathbf{s}_\Theta^{(2)}(\mathbf{x}_t, t) - \mathbf{s}_\phi^{(1)}(\mathbf{x}_t, t)$ (notice that the sign gets flipped). However, we find that aligning the reward function gradients to the difference in means $\boldsymbol{\mu}_\phi(\mathbf{x}_t, t) - \boldsymbol{\mu}_\Theta(\mathbf{x}_t, t)$ (instead of the difference in scores) leads to slightly better performance. Hence, in practice, we use the following reward function:

$$L_{\text{RRF}}(\theta) = \mathbb{E}_{t\sim\mathcal{U}[0,T],\mathbf{x}_t\sim p_t}\left[||\nabla_{\mathbf{x}}\rho_\theta(\mathbf{x}_t, t) - (\boldsymbol{\mu}_\phi(\mathbf{x}_t, t) - \boldsymbol{\mu}_\Theta(\mathbf{x}_t, t))||_2^2\right] \tag{45}$$

Notice that, since during sampling we multiply the outputs of $\rho$ by a guidance scale $\omega > 0$ of our choice, this reward function is equivalent to the one in the main paper, up to a scale factor.

Table 4: Model architectures and dimensions of the relative reward functions.

| Environment | Architecture | Dimensions | Diffusion Horizon | Reward Horizon |
|---|---|---|---|---|
| maze2d-open-v0 | MLP | (32,) | 256 | 64 |
| maze2d-umaze-v0 | MLP | (128,64,32) | 256 | 4 |
| maze2d-medium-v1 | MLP | (128,64,32) | 256 | 4 |
| maze2d-large-v1 | MLP | (128,64,32) | 256 | 4 |
| Halfcheetah | UNet Encoder | (64,64,128,128) | 4 | 4 |
| Hopper | UNet Encoder | (64,64,128,128) | 32 | 32 |
| Walker2D | UNet Encoder | (64,64,128,128) | 32 | 32 |
| Stable Diffusion | UNet Encoder | (32,64,128,256) | - | - |

Table 5: Parameters for the training of the relative reward functions.

| Environment | Learning Rate | Batch Size | Diffusion Timesteps | Training Steps |
|---|---|---|---|---|
| Maze | $5 \cdot 10^{-5}$ | 256 | 100 | 100000 |
| Locomotion | $2 \cdot 10^{-4}$ | 128 | 20 | 50000 |
| Stable Diffusion | $1 \cdot 10^{-4}$ | 64 | 50 | 6000 |

Table 6: Parameters for sampling (rollouts) in Locomotion – guidance scale $\omega$.

| Environment | Discriminator | Reward (Ours, Ablation) | Reward (Ours) |
|---|---|---|---|
| Halfcheetah | $\omega = 0.5$ | $\omega = 0.1$ | $\omega = 0.2$ |
| Hopper | $\omega = 0.4$ | $\omega = 0.1$ | $\omega = 0.4$ |
| Walker2d | $\omega = 0.2$ | $\omega = 0.3$ | $\omega = 0.3$ |

### E.1.6   Trajectory Normalization.

Following the implementation of Janner et al. [25], we use min-max normalization for trajectories in Maze and Locomotion environments. The minima and maxima are computed for each coordinate of states and actions, across all transitions in the dataset. Importantly, they are computed without any noise being added to the transitions. For our experiments with Stable Diffusion, we use standard scaling (i.e. making the data be centered at **0** and have variance 1 on each coordinate). Since in this case a dataset is pre-generated containing all intermediate diffusion timesteps, the means and standard deviations are computed separately for each diffusion timestep, and for each coordinate of the CLIP latents.

### E.1.7   Reward Function Horizon vs. Diffuser Horizon.

We consider different ways of picking the number of timesteps fed into our network for reward prediction. Note that the horizon of the reward model network must be compatible with the length of the Diffuser trajectories (256). To account for this, we choose horizons $h$ that are powers of two and split a given diffusor trajectory of length 256 into $256/h$ sub-trajectories, for which we compute the reward separately. We then average the results of all windows (instead of summing them, so that the final numerical value does not scale inversely with the chosen horizon). The reward horizon $h$ used for different experiments is indicated in Table 4.

### E.1.8   Visualization Details.

To obtain visualizations of the reward function that are intuitively interpretable, we modified the D4RL [12] dataset in Maze2D to use start and goal positions uniformly distributed in space, as opposed to only being supported at integer coordinates. Note that we found that this modification had no influence on performance, but was only made to achieve visually interpretable reward functions (as they are shown in Figure 3).

### E.1.9   Quantitative Evaluation in Maze2D.

We quantitatively evaluated the learned reward functions in Maze2D by comparing the position of the global maximum of the learned reward function to the true goal position. We first smoothed the reward functions slightly to remove artifacts and verified whether the position of the global reward maximum was within 1 grid cell of the correct goal position. We allow for this margin as in the trajectory generation procedure the goal position is not given as an exact position but by a circle with a diameter of 50% of a cell's side length.

We train reward functions for 8 distinct goal positions in each of the four maze configurations and for 5 seeds each. The confidence intervals are computed as $\hat{p} \pm 1.96\sqrt{\frac{\hat{p}\,(1-\hat{p})}{n}}$, where $\hat{p}$ is the fraction of correct goal predictions, and $n$ is the total number of predictions (e.g. for the total accuracy estimate, $n = 4$ mazes $\cdot$ 8 goals $\cdot$ 5 seeds $= 160$).

### E.1.10   Running Time.

The Diffusers [25] take approximately 24 hours to train for 1 million steps on an NVIDIA P40 GPU. Reward functions took around 2 hours to train for 100000 steps for Maze environments, and around 4.5 hours for 50000 steps in Locomotion environments, also on an NVIDIA Tesla P40 GPU. For the Stable Diffusion experiment, it took around 50 minutes to run 6000 training steps on an NVIDIA Tesla M40 GPU. For Locomotion environment evaluations, we ran 512 independent rollouts in parallel for each steered model. Running all 512 of them took around 1 hour, also on an NVIDIA Tesla P40.

Table 7: Parameters for sampling (rollouts) in Locomotion – `t_stopgrad`.

| Environment | Discriminator | Reward (Ours, Ablation) | Reward (Ours) |
|---|---|---|---|
| Halfcheetah | `t_stopgrad = 8` | `t_stopgrad = 0` | `t_stopgrad = 1` |
| Hopper | `t_stopgrad = 10` | `t_stopgrad = 2` | `t_stopgrad = 1` |
| Walker2d | `t_stopgrad = 1` | `t_stopgrad = 5` | `t_stopgrad = 2` |

Table 8: Results for ablation on generalization in Locomotion.

| Environment | Unsteered | Reward (Ours, Ablation) |
|---|---|---|
| Halfcheetah | $31.74 \pm 0.37$ | $33.06 \pm 0.31$ |
| Hopper | $22.95 \pm 0.81$ | $25.03 \pm 0.79$ |
| Walker2d | $30.44 \pm 1.01$ | $42.40 \pm 1.07$ |
| Mean | 28.38 | 33.50 |

## E.2    Additional Results.

### E.2.1    Retraining of Policies in Maze2D with the Extracted Reward Function.

We conducted additional experiments that show that using our extracted reward function, agents can be trained from scratch in Maze2D. We observe in table 2 that agents trained this way with PPO [60] achieve 73.72% of the rewards compared to those trained with the ground-truth reward function, underscoring the robustness of our extracted function; not that a randomly acting policy achieves close to zero performance.

## E.3    Ablations

### E.3.1    Ablation study on dataset size in Maze2D

The main experiments in Maze2D were conducted with datasets of 10 million transitions. To evaluate the sensitivity of our method to dataset size, we conducted a small ablation study of 24 configurations with datasets that contained only 10 thousand transitions, hence on the order of tens of trajectories. The accuracy in this ablation was at 75.0% (as compared to 78.12% in the main experiments). Examples of visual results for the learned reward functions are shown in Figur 6. This ablation is of high relevance, as it indicates that our method can achieve good performance also from little data.

### E.3.2    Ablation Study on Generalization in Locomotion

For the main steering experiments in Locomotion the reward functions are trained on the same base diffusion model that is then steered after. We conducted an ablation study to investigate whether the learned reward

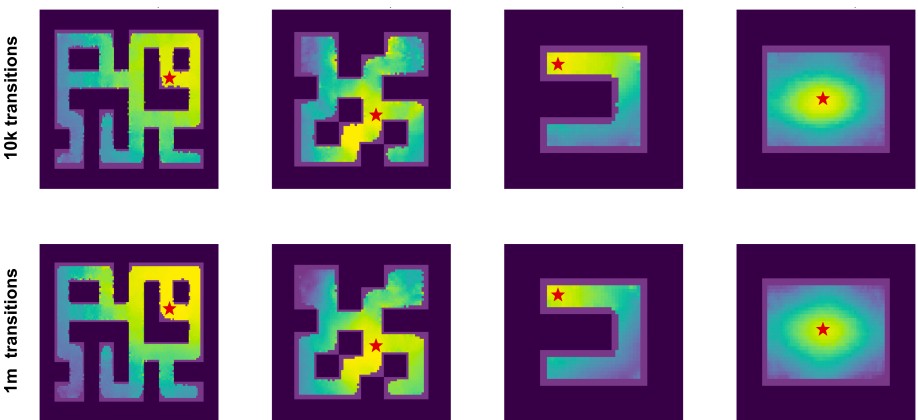

Figure 6: Heatmaps for rewards learned with 10 thousand and 1 million transition datasets. While there is slight performance degradation, the maxima of the reward still largely correspond to the ground-truth goals.

function generalizes to *other base* models, i.e. yields significant performance increases when used to steer a base model that was not part of the training process. We trained additional base models with new seeds and steered these base diffusion models with the previously learned reward function. We report results for this ablation in Table 8. We found that the relative increase in performance is similar to that reported in the main results in Table 1 and therefore conclude that our learned reward function generalizes to new base diffusion models.

### E.3.3 Ablation Study on Hyerparameters in Locomotion.

We conducted additional ablations with respect to t-stopgrad and the guide scale in the Locomotion environments, for a decreased and an increased value each (hence four additional experiments per environment). We observe in Table 3 that results are stable across different parameters.

# Appendix F  Additional Results for Maze2D

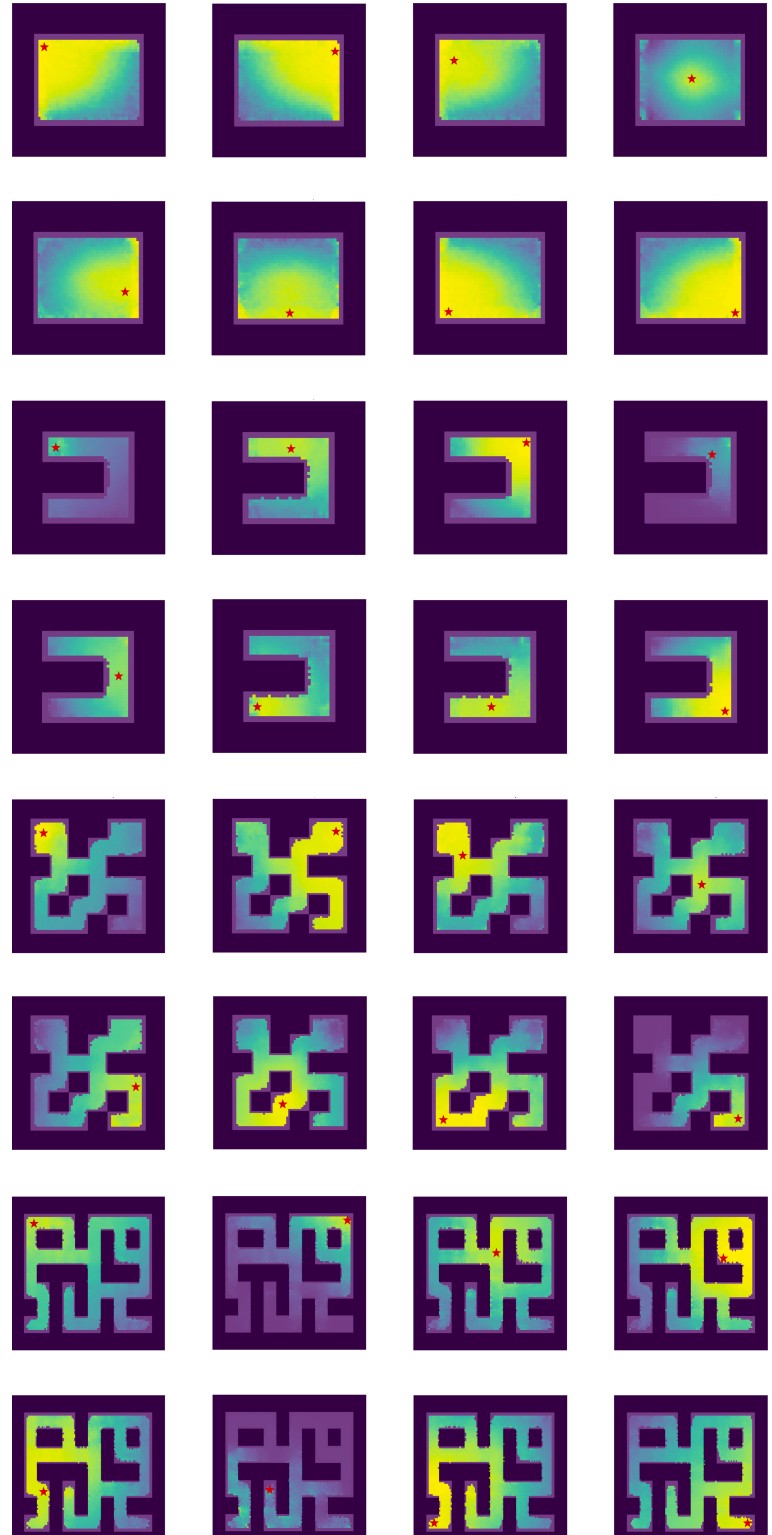

