# OpenReview forum: "Extracting Reward Functions from Diffusion Models"
_NeurIPS.cc/2023/Conference — NeurIPS 2023 poster_

### Official Review · Reviewer_rtGf · 2023-07-01

**Soundness:** 4 excellent
**Presentation:** 3 good
**Contribution:** 3 good
**Rating:** 7
**Confidence:** 3

**Summary:**

The study focuses on extracting a reward function by comparing low-reward and high-reward decision-making diffusion models, sharing the similar goal with inverse reinforcement learning. The researchers define a relative reward function of two diffusion models and establish its existence and uniqueness under certain conditions. They develop a practical learning algorithm to extract the reward function by aligning neural network-parametrized reward function gradients with the difference in outputs from both diffusion models. The method successfully finds correct reward functions in navigation environments and improves performance in standard locomotion benchmarks when steering the base model. Furthermore, the approach generalizes beyond sequential decision-making by learning a reward-like function from two large-scale image generation diffusion models, effectively assigning lower rewards to harmful images.

**Strengths:**

1. The research problem tackled in this study is both significant and clearly defined. Its importance and relevance are well-recognized within the research community.

2. The paper features a reader-friendly format, with technical details coherently presented and logically interconnected.

3. The theoretical findings are fundamental and grounded in reasonable assumptions. Intuitive examples are provided to facilitate practical implementation.

4. Promising empirical results are demonstrated across various tasks, including gridworld, locomotion benchmarks, and image generation.

**Weaknesses:**

One of my primary concerns lies in the connection to inverse reinforcement learning (IRL). Although the paper uses IRL to motivate the task, the relationship to commonly studied IRL methods has not been sufficiently introduced.

1. IRL algorithms typically formulate the problem as a two-player minimax game, assuming bounded rationality of expert data through a maximum entropy framework. In stochastic environments, these methods often analyze causal entropy. However, it appears that such methods have not been thoroughly compared or investigated in this work.

2. In IRL, researchers typically have access to an environment for interaction, which allows for exploration and the generation of nominal trajectories. In the context of diffusion models, there seems to be no such interaction, or at least none observed within the algorithm. Consequently, this paper may be more akin to inverse optimal control, where preference learning is commonly implemented offline without any interaction.

Despite the aforementioned concerns, the overall quality and contributions of this work remain substantial and noteworthy.

**Questions:**

I am wondering whether the diffuision models can achieve comparable performance with popular RL methods like SAC, PPO or DDPG.

**Limitations:**

None.

---

> ### Author Rebuttal · Authors · 2023-08-09
>
> **Weaknesses/ comments:**
> >IRL algorithms typically formulate the problem as a two-player minimax game, assuming bounded rationality of expert data through a maximum entropy framework. In stochastic environments, these methods often analyze causal entropy. However, it appears that such methods have not been thoroughly compared or investigated in this work.
>
> R1: As pointed out by the reviewer in the next question, IRL methods typically rely on environment access and access to a set of expert demonstrations. In contrast, our method does not require environment access, as it assumes that the expert behavior is modeled by a pre-trained diffusion model. We have now clarified this in the respective parts of the introduction and related work sections.
>
> We have now also added an experimental comparison with IRL, benchmarking against the classic AIRL [B] method in Maze2D. To ensure a fair comparison, we ran a grid search over 4 values each for four relevant hyperparameters. We find that our method outperforms AIRL by a clear margin (see table at the bottom of this response). We would however like to point out that we do not see IRL as a comparable baseline, as it makes different assumptions.
>
> > In IRL, researchers typically have access to an environment for interaction, which allows for exploration and the generation of nominal trajectories. In the context of diffusion models, there seems to be no such interaction, or at least none observed within the algorithm. Consequently, this paper may be more akin to inverse optimal control, where preference learning is commonly implemented offline without any interaction.
>
> R2: We agree that there are some parallels to IOC. However, we use comparisons to IRL mostly to motivate the problem of reward function extraction. We have now stated this more clearly in the introduction, also referencing IOC [C] as a related problem setting.
>
>
> **Questions:**
>
> >I am wondering whether the diffuision models can achieve comparable performance with popular RL methods like SAC, PPO or DDPG.
>
> R3: In accordance with [A], we found that our trained expert diffusion models achieve similar performance to RL agents trained with PPO from scratch in both Maze2D and Locomotion environments.
>
> We further conducted additional experiments in Maze2D, where we retrained RL agents from scratch with PPO [D], using the reward functions learned with our method. We found that these agents achieve 72.73% of the performance of PPO agents trained with the ground truth reward function.
>
> Average performance of agents trained with different reward functions (note that a random policy obtains near zero reward):
> | Environment | Groundtruth Reward  | Relative Reward (Ours) | AIRL     |
> |-------------|---------------      |------------------------|----------|
> | OpenMaze    | 92.89 ± 11.79   | 76.45 ± 19.10      | 53.42 ± 33.75   |
> | UMaze       | 94.94 ± 8.89    | 74.52 ± 18.32      | 69.62 ± 29.75     |
> | MediumMaze  | 423.21 ± 51.30  | 276.10 ± 65.21     | 175.49 ± 133.79    |
> | LargeMaze   | 388.76 ± 121.39 | 267.56 ± 98.45     | 139.59 ± 137.79    |
>
>
>
> **Citations:**
>
> [A] ”Planning with Diffusion for Flexible Behavior Synthesis”, Janner et al., ICML 2022
> [B] "Learning Robust Rewards with Adversarial Inverse Reinforcement Learning", Fu et al., ICLR 2018
> [C] "From inverse optimal control to inverse reinforcement learning: A historical review", ab Azar et al., Annual Reviews in Control 2020
> [D] "Proximal Policy Optimization", Schulman et al. 2017

---

> > ### Comment · Reviewer_rtGf · 2023-08-16
> >
> > Thanks to the Authors for the clarification. I will keep my score as is.

---

### Official Review · Reviewer_NVyM · 2023-07-06

**Soundness:** 4 excellent
**Presentation:** 4 excellent
**Contribution:** 2 fair
**Rating:** 7
**Confidence:** 3

**Summary:**

The authors propose a new problem setting, similar to IRL, of extracting reward functions from decision-making diffusion models: one expert and one base (non-expert), where the reward is defined as a function that can steer the base model to the expert through classifier guidance.  Using this intuition, they derive a relative reward function of two diffusion models as an approximation to this steering function and provide theoretical evidence for its uniqueness.

They test this method on maze navigation, locomotion, and image generation tasks and provide a variety of analysis that the learned reward function is reasonable.  Furthermore, they demonstrate that guiding the non-expert model with the extracted reward function does indeed increase task performance in the locomotion tasks.


**Strengths:**

- The authors propose a novel problem of extracting reward functions from decision-making diffusion models.
- The method is simple and practical with sound derivation and some theoretical backing.
- Good range of qualitative and quantitative results across different domains in control and image generation.  Overall, the proposed method seems to be able to extract reasonable reward functions.


**Weaknesses:**

- The motivation for the problem setting is not obvious.  Since you have access to the expert model already, extracting a reward function is not useful for task learning like in inverse RL cases.  Further discussion or demonstrations of use cases for such a reward function would be helpful.
- The experiments lack proper evaluation on classifier guidance except for the locomotion tasks.  As stated in the problem setting, the objective is to extract the reward function that can steer the base model to the expert model through classifier guidance, so this evaluation should be included in more domains.


**Questions:**

- For the locomotion experiments, what do the expert policy rewards look like?  Does your method represent a significant improvement to the policy?  It would also be helpful to have an upper bound figure using a ground truth classifier.
- The problem setting, motivation, and evaluation criteria are all a bit unclear.  Defining the problem explicitly (what makes a good reward function?), with convincing use cases, and bringing the experiments in line would make this a strong paper.


**Limitations:**

Yes, they stated no real world experiments.

---

> ### Author Rebuttal · Authors · 2023-08-09
>
> **Weaknesses/ Notes**
> > The experiments lack proper evaluation on classifier guidance except for the locomotion tasks. As stated in the problem setting, the objective is to extract the reward function that can steer the base model to the expert model through classifier guidance, so this evaluation should be included in more domains.
>
> R1: We would like to point out that the ultimate goal is to learn the relative reward function, which we argue is equivalent to finding the reward function that would allow for "most closely" steering the base model into the expert model.
>
> As the low-dimensional state of Maze2D allows a quantitative and qualitative evaluation of the learned reward functions (in contrast to the high-dimensional Locomotion environments), we directly evaluate the reward functions instead of using them to steer the base model.
>
> We have now conducted additional experiments that demonstrate that the learned reward function in Maze2D can be used to train agents from scratch with PPO [A], achieving 73.72% of the performance of an agent trained with the ground truth reward function (see table below).
>
> Average performance of agents trained with different reward functions (note that a random policy obtains near zero reward):
> | Environment | Groundtruth Reward  | Relative Reward (Ours) |
> |-------------|---------------      |------------------------|
> | OpenMaze    | 92.89 ± 11.79   | 76.45 ± 19.10      |
> | UMaze       | 94.94 ± 8.89    | 74.52 ± 18.32      |
> | MediumMaze  | 423.21 ± 51.30  | 276.10 ± 65.21     |
> | LargeMaze   | 388.76 ± 121.39 | 267.56 ± 98.45     |
>
>
> **Questions**
>
> >For the locomotion experiments, what do the expert policy rewards look like? Does your method represent a significant improvement to the policy? It would also be helpful to have an upper bound figure using a ground truth classifier.
>
> R2: The expert policy consistenty achieves > 95% across all three Locomotion environments.
>
> We have now conducted additional experiments in which we steer a medium-performance diffusion model (trained on the medium-expert D4RL datasets) with the learned relative reward function; see table below. We find that our learned reward function significantly improves performance, by 7.34 on average, which is even larger than the performance increase of 4.61 found in the experiments in the main paper.
>
> Performance in unsteered and RRF steered scenario for medium-performance base diffusion model:
> | Environment | Unsteered     | Relative Reward (Ours)     |
> |-------------|---------------|-------------------|
> | Halfcheetah | 59.41 ± 0.87  | 69.32 ± 0.80   |
> | Hopper      | 58.80 ± 1.01  | 64.97 ± 1.15   |
> | Walker2d    | 96.12 ± 0.92  | 102.05 ± 1.15   |
> | Mean        | 71.44         | 78.78         |
>
> >The problem setting, motivation, and evaluation criteria are all a bit unclear. Defining the problem explicitly (what makes a good reward function?), with convincing use cases, and bringing the experiments in line would make this a strong paper.
>
> R3: The problem statement, broadly speaking, is to learn a reward function that quantifies the differences between two diffusion models. When one of these is taken to model a more "general" distribution and the other a more narrow distribution, we can interpret the latter as an expert model, and the relative reward then encodes the objective that the expert is trying to achieve.
>
> In the Maze2D experiments, we qualitatively evaluate the learned reward functions in Figures 3 and 6. We now also added an experiment demonstrating that the learned reward function allows to retrain agents from scratch (achieving 73.72% of the ground truth performance, see table below). In the high-dimensional Locomotion experiments we evaluate the learned reward function by steering a lower-performanct base model, and now also have additional experiments in this case (as outlined above). In the large-scale Stable Diffusion experiment, the difference between the base and expert models is that the latter is tailored to not producing inappropriate images, which is what we observe when evaluating the learned reward function, which penalizes inappropriate images.
>
> Two potential use cases our work attempts to demonstrate are:
> (1) Extracting reward functions from decision-making diffusion models: Obtaining a reward function allows for interpreting behavior differences, for composition and manipulation of reward functions, and allows to either train agents from scratch or fine-tune existing policies.
> (2) Better understanding diffusion models by contrasting them: The biases of large models trained on different datasets are not always obvious, and our method may aid interpretability and auditing of models by revealing the differences between the outputs they are producing.
>
> Average performance of agents trained with different reward functions (note that a random policy obtains near zero reward):
> | Environment | Groundtruth Reward  | Relative Reward (Ours) |
> |-------------|---------------      |------------------------|
> | OpenMaze    | 92.89 ± 11.79   | 76.45 ± 19.10      |
> | UMaze       | 94.94 ± 8.89    | 74.52 ± 18.32      |
> | MediumMaze  | 423.21 ± 51.30  | 276.10 ± 65.21     |
> | LargeMaze   | 388.76 ± 121.39 | 267.56 ± 98.45     |
>
> **Citations**
> [A] "Proximal Policy Optimization", Schulman et al. 2017

---

> > ### Comment · Reviewer_NVyM · 2023-08-16
> >
> > Thank you for your response and additional experiments. This has addressed all my concerns well.

---

### Official Review · Reviewer_TdMi · 2023-07-11

**Soundness:** 2 fair
**Presentation:** 2 fair
**Contribution:** 2 fair
**Rating:** 4
**Confidence:** 2

**Summary:**

This paper proposed a reward extraction method for reinforcement learning from diffusion models.
Theoretical justification and experiments in diverse domains and settings were used to support the proposed method.

I have no background in reinforcement learning and I think I'm not qualified to evaluate the contribution & novelty of this work. I will be stick with feedback from other reviewers and change my rating appropriately.

**Strengths:**

* Theoretical analysis was provided to justify the proposed method;
* Experiments under diverse settings and domains endorse the effectiveness of the proposed method.

**Weaknesses:**

* What's the benefit of using diffusion models rather than other probabilistic generative models, e.g. VAE, FLOW and GAN, for reward function extraction?
* The drawback of using diffusion models is not mentioned. Will diffusion models slowdown the reward function calculation because iterative denoising is super time-consuming. Can the proposed method be applied to realtime applications?

**Questions:**

See weekness

---

> ### Author Rebuttal · Authors · 2023-08-09
>
> **Weaknesses/ Questions**
>
> >What's the benefit of using diffusion models rather than other probabilistic generative models, e.g. VAE, FLOW and GAN, for reward function extraction?
>
> R1: Diffusion models produce samples by gradual denoising, as opposed to VAEs and GANs. Thus, they can be seen (informally speaking) as a vector field continuously steering the denoised samples. This interpretation then allows us to extract a relative reward by considering the difference of these vector fields. Such an approach would not be feasible with VAEs or GANs.
>
> Furthermore, diffusion models have recently been successfully applied to large-scale generative modeling, as well as decision making, allowing for broad and interesting applications of any methods focused on them.
>
> >The drawback of using diffusion models is not mentioned. Will diffusion models slowdown the reward function calculation because iterative denoising is super time-consuming. Can the proposed method be applied to real-time applications?
>
> R2: The learned reward function is a feed-forward network, and not a diffusion model. Hence, it has much faster inference and allows real-time applications. Furthermore, even though diffusion-based policies may be slower due to the denoising process, this does not slow down learning the relative reward function, as all denoising steps are utilized (and not only the final denoised sample), such that the whole denoising process does not have to be run at each training step.
>
> **Conclusion**:
> Given these clarifications and in light of the other reviews, we wanted to kindly ask the reviewer to consider updating the score. Please let us know of any further questions.

---

> ### Author Response · Authors · 2023-08-17
> **Friendly reminder -- please respond to our rebuttal**
>
> Dear Reviewer,
> Thank you again for your review. We hope that our rebuttal addresses your main concerns. We would greatly appreciate your feedback and would like to know if there are any open questions.

---

### Official Review · Reviewer_rnSj · 2023-07-18

**Soundness:** 3 good
**Presentation:** 3 good
**Contribution:** 3 good
**Rating:** 5
**Confidence:** 3

**Summary:**

This paper proposed a method for extracting reward functions from diffusion models, which can be used for sequential decision-making. The authors defined the notion of a relative reward function of two diffusion models and show conditions under which it exists and is unique. They also propose a practical learning algorithm for finding the relative reward function by aligning the gradients of a neural network to the difference in outputs of the base and expert diffusion models. They demonstrate the results on several domains, such as navigation, locomotion, and image generation, and show that it can recover the correct reward functions and improve the performance of the base models.

**Strengths:**

The paper provides a thorough analysis and empirical evaluation of its approach. It also provides a clear explanation of the proposed method and its relation to previous work. It proposes a new method for extracting reward functions from diffusion models that does not require environment access, simulators, or iterative policy optimization. It is also agnostic to the architecture of the diffusion models used, being applicable to continuous and discrete models, and making no assumption of whether the models are unguided or whether they use either classifier guidance or classifier-free guidance.

**Weaknesses:**

1. The paper assumes that the two diffusion models have the same architecture and are trained on the same data distribution. This may limit the applicability of the method to scenarios where the diffusion models have different architectures or are trained on different data distributions.
2. The paper does not provide any ablation studies on the hyperparameters of the proposed method. The paper does not provide any qualitative analysis or visualization of the learned reward functions.


**Questions:**

Can this method be used to reduce images with bad quality in the diffusion process? How?

---

> ### Author Rebuttal · Authors · 2023-08-09
>
>
> **Notes/ Weaknesses mentioned:**
>
> > The paper assumes that the two diffusion models have the same architecture and are trained on the same data distribution. This may limit the applicability of the method to scenarios where the diffusion models have different architectures or are trained on different data distributions.
>
> R1: This is actually not true -- our method does not assume that the models have the same architecture, as we point out in L45-L48, and the two models must be trained on different distributions. The only requirement is that both have the same output dimension. Applying our method to scenarios where the base and expert model have different output dimensions, however, would constitute an interesting direction for future research.
>
> > The paper does not provide any ablation studies on the hyperparameters of the proposed method.
>
> R2: We already provide some ablations, with respect to the size of the dataset used to train the diffusion models in appendix F.0.1. Due to computational constraints, we did not run additional ablation studies with respect to all hyperparameters (note that for Maze2D we trained 180 Diffusion models and 30 for Locomotion environments). We now conducted additional ablations with respect to t_stopgrad and the guide scale in the Locomotion environments, for a decreased and an increased value each (hence four additional experiments per environment), compared to the optimal values reported in the paper. We have added these experiments to the appendix and report the lower bound, mean and upper bound in the table below, which indicate robustness of our results.
>
> Performance bounds for ablation of t-stopgrad and guide scales:
> | Environment | Lower bound   | Mean           | Upper bound   |
> |-------------|---------------|----------------|---------------|
> | Halfcheetah | 30.24 ± 0.39        | 30.62          |31.5 ± 0.35          |
> | Hopper      | 19.65 ± 0.27        | 22.32          |25.03 ± 0.65         |
> | Walker2d    | 31.65 ± 0.92        | 34.35          |38.14 ± 1.08         |
>
>
> We further conducted additional experiments in the Locomotion domains in which we steer a **new, unseen** (during training) medium-performance diffusion model (trained using the medium-expert D4RL datasets) using the relative reward function learned from the base diffusion model and the expert diffusion model. We find that our learned reward function significantly improves performance, by 7.34 on average, which is even larger than the performance increase of 4.61 found in the experiments in the main paper, underlining the robustness of our method.
>
> Performance in unsteered and RRF steered scenario for medium-performance base diffusion model:
> | Environment | Unsteered     | Relative Reward (Ours)     |
> |-------------|---------------|-------------------|
> | Halfcheetah | 59.41 ± 0.87  | 69.32 ± 0.80      |
> | Hopper      | 58.80 ± 1.01  | 64.97 ± 1.15      |
> | Walker2d    | 96.12 ± 0.92  | 102.05 ± 1.15     |
> | Mean        | 71.44         | 78.78             |
>
>
> > The paper does not provide any qualitative analysis or visualization of the learned reward functions.
>
> R3: Figures 3 and 6 already show a qualitative analysis of the learned reward functions for Maze2D. The visualized points are colored by the learned reward value.
>
> Note that such a visualization is not possible in the Locomotion environments due to their high dimensionality (in which case we instead measure performance improvements when steering the base models).
>
>
> **Questions:**
>
> > Can this method be used to reduce images with bad quality in the diffusion process? How?
>
> R4: We have not considered this application yet. We believe that a general “image quality improvement reward function” might in principle be learnable from two diffusion models that produce low and high-quality images respectively. This reward function could then be used to improve the image quality of other, potentially more domain-specific image-generation diffusion models. This constitutes an interesting direction for future work.
>
>
> **Conclusion:**
> We would like to politely ask the reviewer to consider updating the score, given the additional experiments that demonstrate the robustness of our method and considering the qualitative analysis presented in the paper. We are happy to answer any additional questions.

---

> > ### Comment · Reviewer_rnSj · 2023-08-17
> >
> > Thanks for the response! I will keep my score.

---

> ### Author Response · Authors · 2023-08-17
> **Friendly reminder -- please respond to our rebuttal**
>
> Dear Reviewer,
> Thank you again for your review. We hope that our rebuttal addresses your main concerns. We would greatly appreciate your feedback and would like to know if there are any open questions.

---

### Official Review · Reviewer_qcPp · 2023-07-26

**Soundness:** 3 good
**Presentation:** 4 excellent
**Contribution:** 2 fair
**Rating:** 5
**Confidence:** 4

**Summary:**

The paper introduces the following problem: given two diffusion models, a base model and an expert model, extract a guidance function such that the output distribution of the base model steered with the guidance function would resemble the output distribution of the expert model. The authors first formalize this guidance function as a notion of relative reward between the two diffusion models. They provide some theoretical definitions and results: first, that under certain mild conditions, there exists a unique guidance function rho that solves the relative reward problem; second, they provide the closed-form unique reward guidance function in an ideal setting (if given the ground truth score functions); third, in the general setting, in a space of vector fields with a desirable property, there exists a unique minimizer of "distance" to the ground truth relative reward and they call it the optimal relative reward gradient; finally, they define epsilon-relative reward functions as a guidance function that is epsilon-close to the optimal relative reward gradient. The general setting is necessary because we cannot guarantee that diffusion models perfectly capture the ground truth score function or that they are conservative (due to the approximation error). Finally the authors give an algorithm for approximating the relative reward function via L2 loss assuming access to the diffusion models but not to a simulator. The authors find using their learned reward as a classifier can reliably separate base vs expert demonstrations in Maze2D, improve performance in locomotion environments, and mitigate generation of shocking images by StableDiffusion.

**Strengths:**

The authors introduce and define the novel problem of extracting a relative reward function between two diffusion models. There is substantial and rigorous theoretical analysis on its existence and uniqueness, first in an idealized setting and then in the general setting as the minimizer of a distance quantity over a space with a desirable property. The content is presented in a clear organization, introducing complexity step by step starting from the simple but unrealistic ideal setting. The method does not require access to the environment or simulators. The experiments include a diverse set of environments (maze, locomotion, image generation) and a diverse set of evaluation methods (discriminating base vs test, performance improvement, manual examination of images, ...).

**Weaknesses:**

The paper aims to extract a reward function from diffusion models but narrows its scope without explanation or justification to a *relative* reward function between a base model and expert model. It would be nice to see the authors' reasoning on this: do they claim this is an equivalent problem (perhaps if the base model is chosen cleverly)? Is it simply an easier problem than extracting an "absolute" reward function? Or perhaps we use the relative setup but interpret the result as an absolute reward function? (The experiments seem to support this interpretation since they evaluate whether the learned reward improves performance rather than reproduces the expert model distribution.)

The algorithm for approximating the relative reward function is fairly obvious; if we want to guide model 1 to be like model 2, given the simple additive structure of guidance, we can simply "undo" model 1's drift term and add model 2's. (This weakness is balanced by the strength of the rigorous theoretical analysis!)

It seems that the experiments primarily assess directional effect of the reward guidance, i.e. whether guiding with the reward increases performance compared to without guidance. But the paper claims to extract a relative reward; in the idealized setting using the reward guidance should reproduce the exact distributions of the expert model at every denoising step, and in the general setting we should approximate the expert distributions. This is a higher bar than simply showing improved performance and separation of base vs guided distributions. It would be ideal to see results showing that the base model guided with reward is *indistinguishable* from the expert model (i.e. fools a discriminator).

**Questions:**

- It's not clear to me whether the learned per-step reward function can indeed be used as a reward function beyond guiding diffusion models, so it would help to see experiments supporting this claim. Have you explored whether it can be directly used as the reward signal for an RL agent? If so it would help motivate the setting, otherwise it may not be as compelling since we are learning a guidance function that turns one diffusion model into another when we already have access to the expert model.
- Related to the last point under weaknesses, I would like to see results comparing the performance of expert model vs base guided with reward, not just base guided vs base without guidance.
- It would be super interesting to see a comparison of reward learning from this method versus IRL or other classical methods given a common dataset of expert demonstrations / examples. Do you have a hypothesis about the relative strengths of reward learning via diffusion vs RL?
- Is there any way to implement your method using classifier-free guidance?
- What would happen if you guided the expert model with the relative reward? Or an intermediate model that captures some knowledge from expert but not all? Would the reward guidance echo (overweight) the effect of the already-learned features or would it have no effect?
- In the first sentence under section 4 (methods) you mention you establish the connection between the cumulative reward of a trajectory and the value p(y|x) used for classifier guidance, but I either missed it or the connection wasn't made explicit. Is y the optimality variable of the entire trajectory?

**Limitations:**

Yes

---

> ### Author Rebuttal · Authors · 2023-08-09
>
> **Notes / Weaknesses mentioned:**
> > .. explanation or justification to a *relative* reward function ..
>
> R1: The motivation behind the setting is that the *base* model represents general, unguided exploratory behavior (which could intuitively be thought of as a high-entropy exploration policy in the absence of a reward function). In that case, the learned relative reward function can be thought of as an absolute reward function (as shown in Maze2D).
>
> We would further like to point out that learning absolute reward functions is generally ill-posed, as optimal policies are agnostic to certain transformations of the reward function (see Definition 3.3 in [C] for more details).
>
> > .. experiments primarily assess directional effect of the reward guidance ..
>
> R2: Utilizing stochastic gradient descent, we might not reach the global minimum in parameter space. However, we prove that the global minimum of the optimization objective in function space, stated in Equation 14, approximates the relative reward function that "most closely" matches the optimal relative reward gradient.
>
> Further improving the proposed optimisation constitutes an interesting direction for future work. Taking into consideration the new experiments demonstrating that learned reward functions allow training agents from scratch (see below), we believe our method yields good results in practice even with the current procedure.
>
> **Questions:**
>
> > It's not clear to me whether the learned per-step reward function can indeed be used as a reward function beyond guiding diffusion models[...].
>
> R3: We have added an additional evaluation and show that the learned reward functions allow for successfully training agents in Maze2D from scratch, achieving 73.72% of the performance of agents trained with the ground truth reward (see table below).
>
> > I would like to see results comparing the performance of expert model vs base guided with reward.
>
> R4: We would like to point out that we also compare to a baseline where we use Discriminator guidance (see Table 1). We have now also added the performance of the expert model to the main paper (achieving > 95% on all three environments), but do not consider this a comparable baseline as it is not resulting from an extracted reward function.
>
> > It would be super interesting to see a comparison of reward learning from this method versus IRL or other classical methods given a common dataset of expert demonstrations / examples. Do you have a hypothesis about the relative strengths of reward learning via diffusion vs RL?
>
> R3: We've now benchmarked our method against AIRL [B] in Maze2D and also included results from training with PPO [C] using the ground truth reward. For a fair comparison, we ran a grid search on four values each for four different relevant hyperparameters. Our approach notably outperforms AIRL (refer to the table below). However, it is worth noting that IRL assumes both environment access and a dataset of expert demonstrations. In contrast, our method only assumes pre-trained diffusion models without needing environment access.
>
> Average performance of agents trained with different reward functions (note that a random policy obtains near zero reward):
> | Environment | Groundtruth Reward  | Relative Reward (Ours) | AIRL     |
> |-------------|--      |--|--|
> | OpenMaze    | 92.89 ± 11.79   | 76.45 ± 19.10      | 53.42 ± 33.75   |
> | UMaze       | 94.94 ± 8.89    | 74.52 ± 18.32      | 69.62 ± 29.75     |
> | MediumMaze  | 423.21 ± 51.30  | 276.10 ± 65.21     | 175.49 ± 133.79    |
> | LargeMaze   | 388.76 ± 121.39 | 267.56 ± 98.45     | 139.59 ± 137.79    |
>
> > Is there any way to implement your method using classifier-free guidance?
>
> R4: Our method is agnostic to the architecture of the diffusion models, as pointed out in L47-L48. For example, Stable Diffusion uses classifier-free guidance for conditional sampling, and our method is also able to successfully extract rewards from it.
>
> > What would happen if you guided the expert model with the relative reward? Or an intermediate model that captures some knowledge from expert but not all? Would the reward guidance echo (overweight) the effect of the already-learned features or would it have no effect?
>
> R5: We've added the experiments in the table below. Using the learned reward function, the medium-expert model improves by 7.34. The expert model improves by 0.76 (details omitted due to space constraints). This shows that the reward function boosts the performance of varied base models, notably the medium-expert and the base model, without compromising the performance of near-optimal models like the expert.
>
> Performance in unsteered and RRF steered scenario for medium-performance base diffusion model:
> | Environment | Unsteered     | Relative Reward (Ours)     |
> |-------------|---------------|-------------------|
> | Halfcheetah | 59.41 ± 0.87  | 69.32 ± 0.80   |
> | Hopper      | 58.80 ± 1.01  | 64.97 ± 1.15   |
> | Walker2d    | 96.12 ± 0.92  | 102.05 ± 1.15   |
> | Mean        | 71.44         | 78.78         |
>
> > In the first sentence under section 4 (methods) [...] is y the optimality variable of the entire trajectory?
>
> R6: Yes, $y$ corresponds to $\mathcal{O}_{1:T}$, i.e. the vector of optimality variables for all timesteps of the trajectory. This connection is explored in the "Planning with Diffusion" section (L146-L154).
>
> **Conclusion:**
>
> We would like to politely ask whether the reviewer would be willing to update the score given the added experimental results that show performance favorable to AIRL and the ability to utilize the learned reward function to train agents from scratch. We are happy to address any remaining questions.
>
> **Citations:**
> [A] ”Quantifying differences in reward functions”, Gleave et al., ICLR 2021
> [B] "Learning Robust Rewards with Adversarial Inverse Reinforcement Learning", Fu et al., ICLR 2018
> [C] "Proximal Policy Optimization", Schulman et al. 2017

---

> > ### Comment · Reviewer_qcPp · 2023-08-21
> > **Thank you for the responses**
> >
> > Thank you for the explanation that if the base model represents unguided behavior (for example uniform random) then the learned reward is essentially absolute, this has addressed my question. This is a crucial detail that would greatly benefit readers if it were emphasized in the paper.
> >
> > I do not believe the authors adequately addressed my comment that while the paper claims to be able to match the denoising process at every step, the experiments only assess that using the learned reward improves performance. The experiments are a couple steps away from the theory.
> >
> > Thanks for the clarification that the method works for classifier-based and classifier-free guidance, and for pointing out that you already have experiments demonstrating your method on the classifier-free stable diffusion model. I think it would help to mention the fact that the method works for CG and CFG.
> >
> > Thanks also for the additional experiments demonstrating that 1) PPO agents can indeed be trained successfully using the learned reward, 2) that this outperforms AIRL in maze environments, and 3) that the reward can also improve medium-expert models. In 3) I think the small performance boost in the expert model likely is not statistically significant because the reward only contains information the expert model already has.
> >
> > The additional experiments certainly help me better understand the performance of the method, but I agree with reviewer CPFx that the primary concern is the unrealistic requirement of having an expert diffusion model. In order to obtain such a model, you would need to have the reward in hand already, in which case it doesn't make sense to extract a reward back out again. AIRL differs in that it only assumes access to optimal demonstrations, which is more realistic because we might have human experts provide demonstrations, but we don't have an accurate diffusion model of human demonstrations, and if we did then we would have already solved our problem.
> >
> > Overall I think the work is certainly novel, technically sound (the theory is solid), and the authors ran an impressive set of experiments, but the unrealistic requirement of having an expert diffusion model has not been motivated satisfactorily. If this were better addressed or the main contribution of the paper was reframed differently than reward learning (for example interpretability as reviewer CPFx suggests), the strength of this submission would certainly be improved. I will keep my score as is.

---

> > > ### Author Response · Authors · 2023-08-21
> > > **Clarification: an expert diffusion model can be obtained from demonstrations**
> > >
> > > We thank the reviewer for their response. We would like to make a few clarifications:
> > >
> > > > … while the paper claims to be able to match the denoising process at every step …
> > >
> > > We would like to clarify that the paper does not claim to exactly match the denoising process at every step. Instead, we train a  network to do this approximately, and justify why doing this allows us to extract a canonical notion of relative reward. However, it is nowhere claimed in the paper that the learned, approximate reward must allow for exactly matching the denoising process of the expert model when being used to steer the base model.
> > >
> > > > … I think it would help to mention the fact that the method works for CG and CFG.
> > >
> > > We thank the reviewer for the remark about the applicability of our method to both classifier guidance and classifier-free guidance. We consider a strength of our method the fact that it is agnostic to the architecture and sampling method used by the diffusion models involved.
> > >
> > > > … In order to obtain such a model, you would need to have the reward in hand already …
> > > We would like to clarify that the work of Janner et al. presents a way of obtaining an expert diffusion model from purely observational data, without access to the reward function. Their method only uses the ground-truth reward to do classifier guidance on such a diffusion model, but does not use the reward to train the diffusion model itself.
> > >
> > > As such, it is possible to obtain an expert diffusion model only from demonstration, without needing to have the reward at hand. The expert diffusion models used in our Maze2D and Locomotion are obtained in this way: we use the offline datasets from D4RL to, without accessing any ground truth reward, obtain diffusion models reproducing expert behavior.
> > >
> > > > … but the unrealistic requirement of having an expert diffusion model has not been motivated satisfactorily …
> > >
> > > Taking the above into consideration, we believe that the access to the expert diffusion model is not as unrealistic as posed, since such a model can be obtained from optimal demonstrations, as per Janner et al.
> > >
> > > Hence, we would like to ask if the reviewer would consider revising their score, given our clarifications concerning the main points of improvement raised.

---

> > > > ### Comment · Reviewer_qcPp · 2023-08-21
> > > > **Clarification question: diffusion reward learning as alternative to other IRL approaches?**
> > > >
> > > > Thank you for the response. I have two questions regarding the scope and framing of the paper in light of the clarifications.
> > > >
> > > > 1) Based on the paper draft I interpreted the scope of the paper to be focused on the setting where you have an expert diffusion model and wish to extract a relative reward function between the expert and a sub-optimal model. Previous comments clarified that the scope is not limited to strictly relative reward functions because if the base model is uniform random or high entropy then the relative reward w.r.t the expert can be interpreted as absolute. Is this understanding correct?
> > > >
> > > > 2) The paper states "We consider the problem of extracting a reward function by comparing a decision making diffusion model that models low-reward behavior and one that models high-reward behavior; a setting *related* to inverse reinforcement learning" (emphasis mine) Do the authors claim that the setting is actually precisely inverse RL (given demonstrations extract the underlying reward), and that in the maze environments in the experiments, learning a reward by first fitting a diffusion model to expert demos and then applying their method performs better than the standard AIRL approach? Could a possible explanation be that the former optimization problem is easier than the AIRL objective which involves a potentially unstable GAN?
> > > >
> > > > To summarize, is the paper intended to be framed as an alternative method to AIRL or is the scope more narrow?

---

> > > > > ### Author Response · Authors · 2023-08-21
> > > > > **Responses to clarification questions**
> > > > >
> > > > > We thank the reviewer for their questions. Here are our responses:
> > > > >
> > > > > #### **1.**
> > > > > A relative reward function can indeed be intuitively and informally thought of as “essentially absolute” when the base model produces an exploratory behavior. However, we do not claim this holds in a strong sense, as absolute reward functions are only defined up to certain transformations (e.g. scaling by a positive factor), as noted previously.
> > > > >
> > > > > Understanding the relationship between relative reward functions as defined in the paper and absolute reward functions remains an interesting direction for future work.
> > > > >
> > > > > #### **2.**
> > > > >
> > > > > We do not claim the setting is precisely that of inverse RL, but rather that, if one is able to train decision-making diffusion models on demonstrations for a given task, one can apply our methods to these diffusion models to extract a relative reward. The setting of our method, however, is in a sense broader, as the method also applies when one has access only to the diffusion models themselves, as is the case in the Stable Diffusion experiments. Meanwhile, as noted in the rebuttal, AIRL assumes access to an expert dataset and to the environment.
> > > > >
> > > > > While the results mentioned in the rebuttal suggest our method compares favorably to AIRL on the Maze2D environments, this is not a central claim in the paper, nor is the core purpose of our method to be a better alternative to AIRL.
> > > > >
> > > > > The potential explanation mentioned for why our method would outperform AIRL seems plausible, but a more thorough investigation of how their performances compare (in settings where both methods are applicable), and why, is left to future work.
> > > > >
> > > > > #### **Summary**
> > > > >
> > > > > To summarize, the paper is not intended to be framed as an alternative method to AIRL. Rather, the problem settings are distinct, and our method remains applicable when one has access only to the diffusion models, as is the case with the Stable Diffusion experiments, allowing for applications to settings such as interpretability.

---

> ### Author Response · Authors · 2023-08-17
> **Friendly reminder -- please respond to our rebuttal**
>
> Dear Reviewer,
> Thank you again for your review. We hope that our rebuttal addresses your main concerns. We would greatly appreciate your feedback and would like to know if there are any open questions.

---

### Official Review · Reviewer_CPFx · 2023-07-27

**Soundness:** 3 good
**Presentation:** 2 fair
**Contribution:** 2 fair
**Rating:** 4
**Confidence:** 4

**Summary:**

The authors explore the applications for diffusion models in IRL, and state that they provided the first method for extracting relative reward functions (RRF) from a pair of diffusion models, where the models differ in quality/performance.

They propose a simple but intuitive idea: the difference in score function between base and expert models tells us the relative reward function \rho, and we can train a model to predict this \rho directly.

They build off of the work of Janner et al’s Diffuser, which previously introduced the concept of training a separate model to predict the cumulative rewards of trajectory samples

The authors introduce 3 sets of experiments across 5 domains: classification of expert vs. non-expert trajectories on Maze2D using their trained relative reward function (RRF), higher dimensional control problems (e.g. half cheetah) using classifier guidance (steered by RRF), and classification of synthesized images (with pretrained StableDiffusion network vs. SafeStableDiffusion.)

They also provide proofs as to why the RRF is mathematically sound. I did not have time to check the proof in detail.


**Strengths:**

Generally well written, grammatically correct.

Impressive separability between predicted rewards for generic vs. safe image synthesis models (Figure 2).

Shows 36% improvement for Walker2D with steering from RRF.

Analysis on 5 different domains (4 RL domains, and I2P dataset).


**Weaknesses:**

- How useful is “extracting relative reward functions from two diffusion models”? Applications here don’t seem quite compelling, would be interesting to add a section discussing other potential applications.
- How often would we be given a poor-performing diffusion policy and an expert diffusion policy? What are we trying to solve, and why? Would be helpful to discuss the applications further. Would “image appropriateness classification”, given two models, be a marquee application, and why?

- Diffuser already introduce the idea of extracting the cumulative reward function. When is relative reward superior? No discussion of this.

- Limited numbers of baselines – e.g. no comparison to other RRF techniques, or BC, or Janner et al numbers. Janner et al compare with many baselines (BC CQL IQL DT TT MOPO MOReL MBOP vs. their own). No quantifiable comparison with Diffuser?
- No comparison to reward heatmaps learned in other ways, such as using method of Janner et al
- Appendix is unclear about details given about which diffusion framework used, start/end of schedules, etc? Was Karras-style preconditioning used?

- Requires access to the training dataset for marquee algorithm (Algorithm 1) Why is Algo 2 in the appendix, and not in main paper? Seems like the more useful version, than Algo 1?

- Hard to tell whether it is actually unsafe content since it’s blurred out, overblurred to be not so convincing
- Quantitative results and eval are very sparse. Figure 2: Histogram caption should say on which dataset . One qualitative histogram is not so easy to tell about quantitative performance – why no distributional distances measured?
- No comparison with other baselines for the safe image generation experiment.

- Missing an architecture figure with the two models? Need some sort of graphical illustration / system figure showing training vs. inference behavior and which models are being learned (too much text)
- Bit too much background on diffusion, overpowers contribution. I recommend that the proof on page 5 can be moved to appendix. Main algorithm is only shown on page 7, seems it should be on page 3
- Content placement: Figure 2 is very far from where it is discussed (page 7 vs. page 9)
- Line 278 – “peaks occurring at true goal” -> no discussion of within what neighborhood / distance?

- Weird italicization and capitalization: “Physics informed neural networks”

References seem not complete / unpolished / often incorrect attribution
- Keeps citing Sohl-Dickstein [59] for classifier guidance L24, L47, L143, why not cite Dhariwal et al “Diffusion Models Beat GANs on Image Synthesis”? Where does Sohl-Dickstein mention anything about classifier guidance?

- No author listed for “Artificial Neural Networks for Solving Ordinary and Partial Differential Equations” I. E. Lagaris, A. Likas, D. I. Fotiadis
- No 2022 year listed for [31] Karras, T., Aittala, M., Aila, T. and Laine, S. [n.d.], Elucidating the design space of diffusion-based 418 generative models, in ‘Advances in Neural Information Processing Systems’.
- No 2021 year listed for [63] “[63] Song, Y., Sohl-Dickstein, J., Kingma, D. P., Kumar, A., Ermon, S. and Poole, B. [n.d.], Score-based 489 generative modeling through stochastic differential equations, in ‘International Conference on Learning 490 Representations’.”
- Janner et al bib entry is wrong — accepted to ICML 2022 (correct entry is here: https://diffusion-planning.github.io/files/bib.txt)
- Not dated for [21] Ho, J. and Salimans, T. [n.d.], Classifier-free diffusion guidance, in ‘NeurIPS 2021 Workshop on Deep 398 Generative Models and Downstream Applications’.

- L321 incorrect statement “Safe Stable Diffusion [58], a modified version of Stable Diffusion designed to mitigate the generation of shocking images.” -> See Table 1 of https://arxiv.org/pdf/2211.05105.pdf, shocking is just one of 7 categories of inappropriate images. Inappropriate images are the focus

**Questions:**

- Is there any concept of “relative reward function” in existing literature?
- L264 why are 32 expert models required?
- L263 why are 8 expert models required?
- Why is access to the original dataset required for Algorithm 1? In what real-world settings would this be available?
- Is no steering occurring in Section 5.1, on Maze2D?
- L334 why use “the dataset to generate sets of image embeddings rather than actual images”?


**Limitations:**

No, the authors do not discuss any limitations of their work, and they do not discuss any potential negative societal impact. No Neurips checklist is provided.

My understanding was that this checklist is a requirement.

---

> ### Author Rebuttal · Authors · 2023-08-09
>
> **Notes / Weaknesses mentioned:**
>
> > How useful is “extracting relative reward functions from two diffusion models”? Applications here don’t seem quite compelling.
>
> R1: Two potential use cases our work attempts to demonstrate are:
> (1) Extracting reward functions from decision-making diffusion models: Obtaining a reward function allows for interpreting behavior differences, for composition and manipulation of reward functions, and allows to either train agents from scratch or fine-tune existing policies.
> (2) Better understanding diffusion models by contrasting them: The biases of large models trained on different datasets are not always obvious, and our method aids interpretability and auditing of models by revealing the differences between the outputs they are producing.
>
> > Diffuser already introduce the idea of extracting the cumulative reward function
>
> R2: We believe there is a misunderstanding here. The Diffuser paper [A] does not extract reward functions. In fact, the network used in Diffuser to predict the cumulative reward of a trajectory is trained using the ground truth reward. Meanwhile, our method extracts the relative reward function, not assuming access to the ground truth reward.
>
> > Limited numbers of baselines – e.g. no comparison to other RRF techniques, or BC, or Janner et al numbers. Janner et al compare with many baselines [..]. No quantifiable comparison with Diffuser?
>
> R3: We are not aware of any other RRF techniques -- we kindly invite the reviewer to point to any references to consider. Note that the baselines used in Diffuser [A] do not apply -- our problem setting is distinct from that in Diffuser (we extract reward functions, whereas they produce policies).
>
> > No comparison to reward heatmaps learned in other ways, such as using method of Janner et al
>
> R4: We show learned reward maps in Figures 3 and 6. As mentioned above, Janner [A] does not learn reward heatmaps.
>
> **Questions:**
>
> > Is there any concept of “relative reward function” in existing literature?
>
> R5: Generally, learned reward functions are never absolute, as optimal policies are invariant to certain transformations of the reward function (see Definition 3.3 in [C]). Previous work [C] studies how to quantify differences of given reward functions, but does not extract reward functions from demonstrations.
>
> > L264 why are 32 expert models required? L263 why are 8 expert models required?
>
> R6: In the Maze2D experiments, we evaluate our method’s ability to model diverse sets of 8 goals for each maze configuration. As such, for each maze, 8 expert models are needed; one for each goal position. Since there are 4 mazes in total, this gives 32 expert models across all mazes. As we run this for 5 random seeds, this gives a total of 160 expert models.
>
> > Why is access to the original dataset required for Algorithm 1?
>
> R7: Generally, our problem setting assumes that two diffusion models are given. However, training the relative reward function requires input samples. In the Stable Diffusion experiments, we show how to obtain these samples from the pre-trained diffusion models themselves, while in Maze2D and Locomotion we simply use the given datasets.
>
> > Is no steering occurring in Section 5.1, on Maze2D?
>
> R8: In Maze2D's low-dimensional state space, we can effectively evaluate reward functions, unlike in the high-dimensional Locomotion environments. Thus, we chose to directly assess these functions rather than use them to guide the base model.
>
> We conducted additional experiments, which show that using our extracted reward function, agents can be trained from scratch in Maze2D. Agents trained this way with PPO [D] achieve 73.72% of the rewards compared to those trained with the ground-truth reward function, underscoring the robustness of our extracted function.
>
> Average performance of agents trained with different reward functions (note that a random policy obtains near zero reward):
> | Environment | Groundtruth Reward  | Relative Reward (Ours) |
> |-|-|-|
> | OpenMaze    | 92.89 ± 11.79   | 76.45 ± 19.10      |
> | UMaze       | 94.94 ± 8.89    | 74.52 ± 18.32      |
> | MediumMaze  | 423.21 ± 51.30  | 276.10 ± 65.21     |
> | LargeMaze   | 388.76 ± 121.39 | 267.56 ± 98.45     |
>
> > L334 why use “the dataset to generate sets of image embeddings rather than actual images”?
>
> R9: Our method for reward learning leverages the gradual sampling process of diffusion models. In the case of Stable Diffusion and Safe Stable Diffusion, this process happens in latent space, which is why we only need the image embeddings to train our model. This also simplifies the optimization problem, as the embeddings are lower dimensional than the decoded images.
>
> > do not discuss any limitations of their work, and they do not discuss any potential negative societal impact.
>
> R10: Please refer to the Conclusion section, which discusses the limitations. We now added an additional sentence stating that we expect the potential positive effects of enabling learning of relative reward functions of large pre-trained models, e.g., for understanding and alignment of generated outputs, to outweigh the potential negative effects.
>
> > No checklist.
>
> R11: This year, the checklist is on OpenReview, hence not attached to the paper.
>
> **Conclusion:**
> Given the clarifications around the distinction of our method from Janner's work (Diffuser) and the additional experiments that demonstrate that the learned reward function allows training agents from scratch, we kindly ask the reviewer whether it is possible to revise the review score.
>
> **Citations:**
> [A] ”Planning with Diffusion for Flexible Behavior Synthesis”, Janner et al., ICML 2022
> [B] "Safe latent diffusion: Mitigating inappropriate degeneration in diffusion models", Schramowski et al., CVPR 2023
> [C] ”Quantifying differences in reward functions”, Gleave et al., ICLR 2021
> [D] "Proximal Policy Optimization", Schulman et al. 2017

---

> ### Author Response · Authors · 2023-08-17
> **Friendly reminder -- please respond to our rebuttal**
>
> Dear Reviewer,
>
> Thank you again for your review. We hope that our rebuttal addresses your main concerns. We would greatly appreciate your feedback and would like to know if there are any open questions.

---

> > ### Comment · Reviewer_CPFx · 2023-08-18
> > **Thank you for the authors’ responses**
> >
> > Thank you to the authors for their detailed replies and time spent providing clarifications.
> >
> > Thanks also for the clarification regarding the use of reward in Janner et al.
> >
> > While novel and of some interest theoretically, the method appears to be of limited value for the locomotion experiments, as one could simply use the expert model (and it would be preferred). In addition, assumptions introduce limitations, such as reliance on access to an expert model. I see more value on the interpretability side and content moderation side, but I don’t think the applications demonstrated in the paper indicate convincingly useful applications.
> >
> > The paper seems to be focusing on how one can train a new model with the RRF. If you already have the expert model, why would you want to do that, especially if the method doesn’t allow surpassing the performance of the expert/teacher model? In this case it appears the main value is in model interpretability, which I think should instead be the focus of the paper (main contribution and featured in abstract), whereas the locomotion experiments are presented as the marquee results.
> >
> > I don’t believe the authors have responded to my question regarding “ access to the original dataset required for Algorithm 1…In what real-world settings would this be available?”
> >
> > I also believe the authors have not responded to my question regarding: “Keeps citing Sohl-Dickstein [59] for classifier guidance L24, L47, L143, why not cite Dhariwal et al “Diffusion Models Beat GANs on Image Synthesis”? Where does Sohl-Dickstein mention anything about classifier guidance?”
> >
> > I was hoping the authors would have been willing to provide something based on my comment “Missing an architecture figure with the two models? Need some sort of graphical illustration / system figure showing training vs. inference behavior and which models are being learned (too much text)”, but they have not addressed this.
> >
> > I agree with Reviewer qcPp that “comparison of reward learning from this method versus IRL or other classical methods given a common dataset of expert demonstrations / examples” would strengthen the paper, and I appreciate the authors providing those experiments in their rebuttal. If included, that would have strengthened the paper. I think IRL should be discussed more in the related work section (just 1 sentence devoted to this, but it appears to be the conceptual foundation of the work — derive rewards from expert behavior).
> >
> > I also agree with their assessment that RRF is interpreted as an absolute reward function when training new models, but this is not made clear in the text. (Unclear what “medium-expert” is in the authors’ response).
> >
> > The work appears technically correct. Given the interesting theoretical approach, I believe the work and experiments could be revised and make a strong submission in a subsequent conference, but I keep my prior score. Thank you.

---

> > > ### Author Response · Authors · 2023-08-18
> > >
> > > Dear Reviewer,
> > >
> > > Unfortunately the rebuttal was limited to a certain length, which is why we only included the most important responses in it. Below, please find the responses to all other points you mentioned in your original review.
> > >
> > > > Appendix is unclear about details given about which diffusion framework used, start/end of schedules, etc? Was Karras-style preconditioning used?
> > >
> > > Following Janner et al. Section 3, we use the DDPM framework of Ho et al. 2022 with a cosine beta schedule. We do not use preconditioning in the sense of Section 5 of Karras et al. 2022. However, we do clip the the denoised latents $\mathbf{x}_t$ during sampling, and apply scaling to trajectories, as per the Appendix (L711). We will add these details, and release the source code, which should alleviate any reproducibility concerns.
> > >
> > > > Requires access to the training dataset for marquee algorithm (Algorithm 1) Why is Algo 2 in the appendix, and not in main paper? Seems like the more useful version, than Algo 1?
> > >
> > > We placed algorithm 1 in the main paper as it is more frequently used in the experiments. We appreciate your feedback and will place Algorithm 2 in the main paper for the camera-ready version (for which we will have an additional page).
> > >
> > > > Hard to tell whether it is actually unsafe content since it’s blurred out, overblurred to be not so convincing
> > >
> > > We applied a strong blur because even when blurred it can be easy to guess the disturbing content. The images contain violence and hateful symbolism, which we feel should not be imposed on the reader of a scientific paper. Please refer to the original paper [B] for more detailed descriptions of the unsafe (blurred) images.
> > >
> > > > Quantitative results and eval are very sparse. Figure 2: Histogram caption should say on which dataset. One qualitative histogram is not so easy to tell about quantitative performance – why no distributional distances measured?
> > >
> > > Thank you for this remark. We have now computed the Wasserstein 2 distance for the distributions displayed in Figure 2, which evaluates to 17.74. We added this information to the main paper and updated the caption.
> > >
> > > > No comparison with other baselines for the safe image generation experiment.
> > >
> > > We are not aware of relevant baselines in this scenario, but are happy to add such a comparison if pointed to specific baselines.
> > >
> > > > Missing an architecture figure with the two models? Need some sort of graphical illustration / system figure showing training vs. inference behavior and which models are being learned (too much text)
> > >
> > > We have added such a Figure to the paper, highlighting that both models are required for training the reward function and that only the base diffusion model and the learned reward function are used for inference.
> > >
> > > > Bit too much background on diffusion overpowers contribution. I recommend that the proof on page 5 can be moved to appendix. Main algorithm is only shown on page 7, seems it should be on page 3.
> > >
> > > We have made the overview of diffusion models more concise, however, we would prefer keeping the proof on page 5 in the main paper as we consider it central to our contribution, unless there is a strong opinion to change this.
> > >
> > > > Content placement: Figure 2 is very far from where it is discussed (page 7 vs. page 9)
> > >
> > > We have rearranged the Figures accordingly.
> > >
> > > > Line 278 – “peaks occurring at true goal” -> no discussion of within what neighborhood / distance?
> > >
> > > As discussed in Appendix E we used a tolerance of 1 grid cell, similar to the tolerance used in the original environment reward function. We have now stated this more clearly.
> > >
> > > > Weird italicization and capitalization: “Physics informed neural networks
> > >
> > > We have updated the capitalization in accordance with the rest of the paper; thank you for the remark.
> > >
> > > > Keeps citing Sohl-Dickstein [59] for classifier guidance L24, L47, L143, why not cite Dhariwal et al “Diffusion Models Beat GANs on Image Synthesis”? Where does Sohl-Dickstein mention anything about classifier guidance?
> > >
> > > Sohl-Dickstein et al. [59] introduce the classifier guidance equation in Appendix C, Eq. 61, which is why we cited them over Dhariwal et al. However, recognizing the significant contribution of Dhariwal et al. to the practical implementation, we have also included them in the citation.
> > >
> > > > L321 incorrect statement “Safe Stable Diffusion [58], a modified version of Stable Diffusion designed to mitigate the generation of shocking images.” [...] Inappropriate images are the focus.
> > >
> > > Thank you for this remark, we have updated our wording.*

---

> > > > ### Author Response · Authors · 2023-08-18
> > > > **Additional Comments by reviewer — please respond**
> > > >
> > > > Apart from the questions addressed above, we would also like to address the new points raised by the reviewer.
> > > >
> > > > > Contributions lie more in interpretability and this should be stated more clearly
> > > >
> > > > We agree that our work makes relevant contributions to interpretability and will state this more clearly in abstract and introduction. However, we would like to point out that learning rewards goes beyond simply imitating expert behaviour, which is why we believe our work also makes a relevant contribution to reward learning.
> > > >
> > > > > IRL results strengthen the paper
> > > >
> > > > We will include these results in a camera-ready version, for which we are granted an extra page.
> > > >
> > > > > Unclear what medium-expert model is
> > > > We are sorry for the confusion.
> > > >
> > > > This is simply a diffusion model that performs better
> > > >  than the base model, as it is trained on a mix of medium and expert trajectories (see original D4RL datasets for more details).
> > > >
> > > > We kindly ask the reviewer to confirm whether this addresses their concerns.

---

### Author Rebuttal · Authors · 2023-08-09

We thank all reviewers for their time and valuable feedback, and appreciate that reviewers found our work novel and relevant.

We have addressed each reviewer's questions and concerns individually below.

Please let us know if there are any further questions.

Best wishes,
The Authors

---

### Decision · Program_Chairs · 2023-09-21

**Decision:**

Accept (poster)

**Comment:**

Overall, the reception from the reviewers was somewhat mixed. There were some common concerns regarding the problem setting studied. -- the reviewers were concerned that the problem's relationship to Inverse RL was unclear, and also about the purpose of learning a relative reward function was given access to an expert model already. The authors have clarified that the setting is not the same as IRL, and instead the assumption is merely sample access to two different diffusion models. Furthermore, the authors clarified that the goal was not to learn an expert policy using the learned relative reward function, but rather the recovery of the relative reward function itself, which affords both training policies and quantifying the differences between two models. The reviewers thought that the work was technically correct. Given that several reviewers were fairly positive in their assessment, and that the concerns of the reviewers recommending rejection were about the problem formulation and scope (which appears clarified to me), I recommend acceptance.